# Carroll black holes

Florian Ecker[1], Daniel Grumiller[1,5], Jelle Hartong[2], Alfredo Pérez[3,4], Stefan Prohazka[2], and Ricardo Troncoso[3,4]

**1** Institute for Theoretical Physics, TU Wien
Wiedner Hauptstrasse 8-10, A-1040 Vienna, Austria, Europe

**2** School of Mathematics and Maxwell Institute for Mathematical Sciences
University of Edinburgh, Peter Guthrie Tait Road, Edinburgh EH9 3FD, UK

**3** Centro de Estudios Científicos (CECs), Avenida Arturo Prat 514, Valdivia, Chile

**4** Facultad de Ingeniería, Arquitectura y Diseño, Universidad San Sebastián,
sede Valdivia, General Lagos 1163, Valdivia 5110693, Chile

**5** Theoretical Sciences Visiting Program, Okinawa Institute of Science and Technology
Graduate University, Onna, 904-0495, Japan

fecker@hep.itp.tuwien.ac.at, grumil@hep.itp.tuwien.ac.at, jelle.hartong@ed.ac.uk,
alfredo.perez@uss.cl, stefan.prohazka@ed.ac.uk, ricardo.troncoso@uss.cl

## Abstract

Despite the absence of a lightcone structure, some solutions of Carroll gravity show black hole-like behaviour. We define Carroll black holes as solutions of Carroll gravity that exhibit Carroll thermal properties and have a Carroll extremal surface, notions introduced in our work. The latter is a Carroll analogue of a Lorentzian extremal surface. As examples, we discuss the Carroll versions of Schwarzschild, Reissner–Nordström, and BTZ black holes and black hole solutions of generic 1+1 dimensional Carroll dilaton gravity, including Carroll JT and Carroll Witten black holes.

# 1  Introduction

After laying dormant for about half a century [1,2], Carroll symmetries recently attracted considerable attention. Mathematically, Carroll spacetimes are equipped with degenerate (otherwise Euclidean) metrics with a one-dimensional kernel. For example, a four-dimensional (4d) Carroll spacetime has signature $(0, +, +, +)$. Geometrically, Carroll-signature collapses lightcones so that standard Lorentzian notions of horizons are un-

available. Physically, Carroll symmetries arise in a variety of contexts, including ultra-relativistic (speed of light to zero) and near-horizon limits, which explains their topicality, cf. [3] for a review.

Indeed, Carroll symmetries emerge naturally on null surfaces and therefore find their place at null infinity [4], at black hole horizons [5–7], but also at spatial [8] and time-like infinity [9] in the extreme situation that one direction decouples. Due to its restricted mobility, Carrollian physics shares features with fractons [10–13] (more precisely, particles with conserved electric charge and dipole moment) and is relevant in models with so-called "spacetime subsymmetries" [14, 15] (see also [16, 17]). Another condensed matter application of Carroll symmetries is flat bands [18] (the Fourier-transformed statement of collapsed lightcones). Additionally, Carroll symmetries may be relevant for cosmology [19], black hole microstates [20], CFT deformations [21–24], and govern Bjorken flow [25]. Finally, the conformal Carroll algebra is isomorphic to BMS in one higher dimension [4], which led to numerous developments in flat space holography, especially in 2+1 bulk dimensions [26–36] and in 3+1 bulk dimensions [37–40].

Most applications mentioned above use Carroll symmetries as global spacetime symmetries, i.e., at the same level as special relativity uses Poincaré symmetries. Given the success of general relativity as a theory of gravity, which renders spacetime symmetries local, it is tempting to also make Carroll symmetries local [41, 42] and consider theories of Carroll gravity, e.g., two-dimensional (2d) Carroll dilaton gravity [43, 44].

Whether one considers Carroll gravity as a limit of general relativity (see Fig. 1 below for various versions of such a limit) or as an intrinsic gravity model, one quickly faces a conundrum: some of the solutions of these theories behave in several ways like black holes (see, e.g., [17, 45]) and yet, obviously, there cannot be any black holes in Carroll gravity according to their standard definition in terms of an event horizon [46]. This lacuna motivates the present work and fits a longer-term theme of finding a better black hole definition that may apply to quantum theory.

Our main goal is to define entities, which we shall refer to as "Carroll black holes", that arise either as the (magnetic) Carroll limit of black hole solutions of general relativity, like Schwarzschild–Tangherlini [47, 48], or as intrinsic solutions of Carroll gravity, like the Carroll versions of JT [49, 50] or Witten black holes [51–53].

A key entity that we define in our work is a Carroll extremal surface, a Carroll analogue of a Lorentzian extremal surface (see e.g. [54] for their classical and quantum definitions). Lorentzian extremal surfaces arise as bifurcation surfaces on Killing horizons and are thus part of eternal black hole geometries. Despite the absence of horizons in Carroll geometries, the analogue of a bifurcation surface still exists, and we declare it to be a defining property of Carroll black holes. We shall be more precise and detailed in the body of our paper, but for now, a schematic formula that summarizes our main definition is

$$\text{Carroll black hole } = \text{ Carroll extremal surface } + \text{ Carroll thermal properties}\,. \qquad (1)$$

While the concepts developed in our work are independent of the dimension, we get a lot of mileage from considering simple models of Carroll gravity where the solution space is under complete analytic control. Therefore, in a substantial part of our work, we focus on 2d Carroll dilaton gravity models and, as a byproduct, construct their solution spaces.

Busy readers happy to skip the details can continue with Section 6, which discusses many of the main features of this work in 3+1 dimensions.

This paper is organized as follows. In Section 2, we derive all solutions of 2d Carroll dilaton gravity, also addressing different formulations of these models, different orders of limits, and a higher-dimensional perspective. In Section 3, we focus on the Carroll analogues of some thermal properties, mass, temperature, and entropy, highlighting subtleties

with dimensionalities. In Section 4, we introduce a geometric key concept, Carroll extremal surfaces, to define Carroll black holes. In Section 5, we apply our results and definitions to the Carroll JT model, the Carroll limit of Schwarzschild, the Carroll CGHS model, and the Carroll Witten black hole. In Section 6, we elaborate on the 4d perspective of the Carroll–Schwarzschild black hole and the associated wormhole picture. In Section 7, we generalize our results to charged and rotating Carroll black holes, recovering BPS bounds well-known from the Lorentzian case. In Section 8, we conclude with an outlook of some research avenues suggested by our work. For readers confronted for the first time with global and local Carroll symmetries, we review them concisely in Appendix A. Finally, Appendix B describes a map between Lorentzian and Carrollian Poisson-sigma models.

## 2   Actions and solutions of 2d Carroll dilaton gravity

In this Section, we construct all solutions of 2d Carroll dilaton gravity. We start with a review of generic 2d Carroll dilaton gravity in Subsection 2.1, including different formulations of the same theory. In Subsection 2.2, we obtain 2d Carroll dilaton gravity through different limits. In Subsection 2.3, we derive all classical solutions locally, exploiting methods similar to the Lorentzian case and finding similar results, especially a constant and a linear dilaton sector. We also address some global aspects of the solutions that hint already at special loci, which we shall later identify as Carroll extremal surfaces.

### 2.1   Generic 2d Carroll dilaton gravity

Generic 2d Carroll dilaton gravity was constructed in the first-order and Poisson-sigma model (PSM) formulations in [43]. Here, we recall this model and provide further details, in particular, the equations of motion and the second-order formulation.

#### 2.1.1   First-order formulation

Consider the 2d Carroll dilaton gravity bulk action [43]

$$I_{1^{\text{st}}}[\omega,\,\tau,\,e,\,X,\,X_{\text{H}},\,X_{\text{P}}] = \frac{k}{2\pi}\int_{\mathcal{M}}\mathcal{L} \tag{2}$$

on a 2d manifold $\mathcal{M}$ with coupling $k$ and the Lagrange-2-form

$$\mathcal{L} = X\,\mathrm{d}\omega + X_{\text{H}}\left(\mathrm{d}\tau + \omega \wedge e\right) + X_{\text{P}}\,\mathrm{d}e + \mathcal{V}(X,\,X_{\text{H}})\,\tau \wedge e \tag{3}$$

where the potential $\mathcal{V}(X,\,X_{\text{H}})$ is an arbitrary function. The scalar fields are dilaton $X$ and Lagrange multipliers for torsion constraints $X_{\text{H}}$, $X_{\text{P}}$. The 1-forms are spatial einbein $e$, temporal einbein $\tau$ and Carroll boost connection $\omega$. The composite 2-forms are curvature $\Omega = \mathrm{d}\omega$, torsion $T = \mathrm{d}\tau + \omega \wedge e$, and intrinsic torsion $\Theta = \mathrm{d}e$ where the latter is defined as the part of the torsion independent of the boost connection. We summarily refer to the scalar fields as $X^I = (X, X_{\text{H}}, X_{\text{P}})$ and to the 1-forms as $A_I = (\omega, \tau, e)$.

    The Lagrange-2-form (3) [and hence also the action (2)] is invariant under local Carroll boosts (see Appendix A for a summary of Carroll symmetries)

$$\delta_\lambda X = 0 \qquad\qquad \delta_\lambda X_{\text{H}} = 0 \qquad\qquad \delta_\lambda X_{\text{P}} = X_{\text{H}}\,\lambda \tag{4a}$$

$$\delta_\lambda \omega = \mathrm{d}\lambda \qquad\qquad \delta_\lambda \tau = -e\,\lambda \qquad\qquad \delta_\lambda e = 0\,. \tag{4b}$$

The transformations (4) show that the dilaton $X$, the field $X_{\mathrm{H}}$ and the spatial einbein $e$ are Carroll boost invariant, while the 1-form $\omega$ is the Carroll boost connection. The non-invariances of the temporal einbein $\tau$ and the scalar $X_{\mathrm{P}}$ conspire such that the sum of the middle two terms in the Lagrange-2-form is Carroll boost-invariant.

Moreover, the action (2) is invariant under two additional gauge symmetries $\lambda_{\mathrm{H}}$ and $\lambda_{\mathrm{P}}$ (we define $\partial_X := \partial/\partial X$ and $\partial_{\mathrm{H}} := \partial/\partial X_{\mathrm{H}}$)

$$\delta_{\lambda_{\mathrm{H}}} X = 0 \qquad\qquad \delta_{\lambda_{\mathrm{H}}} X_{\mathrm{H}} = 0 \qquad\qquad \delta_{\lambda_{\mathrm{H}}} X_{\mathrm{P}} = \mathcal{V}\,\lambda_{\mathrm{H}} \tag{5a}$$

$$\delta_{\lambda_{\mathrm{H}}} \omega = -(\partial_X \mathcal{V})\,e\,\lambda_{\mathrm{H}} \qquad \delta_{\lambda_{\mathrm{H}}}\tau = \mathrm{d}\lambda_{\mathrm{H}} - (\partial_{\mathrm{H}}\mathcal{V})\,e\,\lambda_{\mathrm{H}} \qquad \delta_{\lambda_{\mathrm{H}}} e = 0 \tag{5b}$$

and

$$\delta_{\lambda_{\mathrm{P}}} X = -X_{\mathrm{H}}\,\lambda_{\mathrm{P}} \qquad \delta_{\lambda_{\mathrm{P}}} X_{\mathrm{H}} = -\mathcal{V}\,\lambda_{\mathrm{P}} \qquad\qquad \delta_{\lambda_{\mathrm{P}}} X_{\mathrm{P}} = 0 \tag{6a}$$

$$\delta_{\lambda_{\mathrm{P}}} \omega = (\partial_X \mathcal{V})\,\tau\lambda_{\mathrm{P}} \qquad \delta_{\lambda_{\mathrm{P}}}\tau = \omega\,\lambda_{\mathrm{P}} + (\partial_{\mathrm{H}}\mathcal{V})\,\tau\lambda_{\mathrm{P}} \qquad \delta_{\lambda_{\mathrm{P}}} e = \mathrm{d}\lambda_{\mathrm{P}} \tag{6b}$$

On-shell they generate diffeomorphisms along a vector field $\xi^\mu$ by virtue of the standard relations[1] $\lambda_{\mathrm{H}} = \tau_\mu\,\xi^\mu$ and $\lambda_{\mathrm{P}} = e_\mu\,\xi^\mu$:

$$\delta_\xi X \approx \xi^\mu\partial_\mu X \qquad\qquad \delta_\xi \omega_\mu \approx \xi^\nu\partial_\nu\omega_\mu + \omega_\nu\partial_\mu\xi^\nu \tag{7a}$$

$$\delta_\xi X_{\mathrm{H}} \approx \xi^\mu\partial_\mu X_{\mathrm{H}} \qquad\qquad \delta_\xi \tau_\mu \approx \xi^\nu\partial_\nu\tau_\mu + \tau_\nu\partial_\mu\xi^\nu \tag{7b}$$

$$\delta_\xi X_{\mathrm{P}} \approx \xi^\mu\partial_\mu X_{\mathrm{P}} \qquad\qquad \delta_\xi e_\mu \approx \xi^\nu\partial_\nu e_\mu + e_\nu\partial_\mu\xi^\nu \tag{7c}$$

The Lie variations above follow from the gauge symmetries together with the Carrollian equations of motion displayed below in (8) ($\approx$ denotes on-shell equivalence).

### 2.1.2   Equations of motion

Varying the action (2) with respect to all fields yields the equations of motion.

| $\delta X$ | Carroll curvature: | $\Omega = \mathrm{d}\omega = -\partial_X\mathcal{V}(X,\,X_{\mathrm{H}})\,\tau\wedge e$ | (8a) |
| $\delta X_{\mathrm{H}}$ | Carroll torsion: | $T = \mathrm{d}\tau + \omega\wedge e = -\partial_{\mathrm{H}}\mathcal{V}(X,\,X_{\mathrm{H}})\,\tau\wedge e$ | (8b) |
| $\delta X_{\mathrm{P}}$ | No intrinsic torsion: | $\Theta = \mathrm{d}e = 0$ | (8c) |
| $\delta\omega$ | Carroll metric: | $\mathrm{d}X + X_{\mathrm{H}}\,e = 0$ | (8d) |
| $\delta\tau$ | Carroll Casimir: | $\mathrm{d}X_{\mathrm{H}} + \mathcal{V}(X,\,X_{\mathrm{H}})\,e = 0$ | (8e) |
| $\delta e$ | Auxiliary field: | $\mathrm{d}X_{\mathrm{P}} - \mathcal{V}(X,\,X_{\mathrm{H}})\,\tau - X_{\mathrm{H}}\,\omega = 0$ | (8f) |

The first equation (8a) determines the Carroll curvature, which generally is non-zero but trivially vanishes whenever the potential is independent of the dilaton field. The second equation (8b) shows that on-shell Carroll torsion vanishes whenever the potential is independent of $X_{\mathrm{H}}$. The third equation (8c) reveals that there is never intrinsic torsion, regardless of how the potential is chosen. The fourth equation (8d) allows algebraically determining the spatial einbein (and hence the Carroll metric) in terms of the Carroll boost invariant scalars, $X$ and $X_{\mathrm{H}}$. The fifth equation (8e) entails a conserved Casimir function, which we shall uncover below when discussing linear dilaton vacua. The final equation (8f) allows determining the auxiliary field $X_{\mathrm{P}}$ in terms of the potential $\mathcal{V}(X,\,X_{\mathrm{H}})$ and the geometric data extracted from the other five equations of motion or, alternatively, if $X_{\mathrm{P}}$ is gauge fixed suitably it provides an algebraic constraint relating $\tau$ and $\omega$,

It can be useful to map solutions of different models to each other if they can be related by suitable Weyl rescalings. Therefore, consider a dilaton-dependent Weyl rescaling, parametrized by $\alpha$, of the Carroll metric

$$e \to \tilde{e} = e\,e^{\alpha(X)} \tag{9}$$

---

[1]Additionally, we need a compensating Carroll boost generated by $\lambda = \omega_\mu\,\xi^\mu$.

that leaves invariant the dilaton, $X \to \tilde{X} = X$. Such Weyl rescalings are compatible with the absence of intrinsic torsion (this would not be the case if the Weyl factor did depend on time). Consistency with Carroll boosts demands that also $\tau$ scales in the same way as $e$,

$$\tau \to \tilde{\tau} = \tau \, e^{\alpha(X)} \,. \tag{10}$$

The Carroll metric equation (8d) implies that $X_{\mathrm{H}}$ transforms inversely to $e$,

$$X_{\mathrm{H}} \to \tilde{X}_{\mathrm{H}} = X_{\mathrm{H}} \, e^{-\alpha(X)} \tag{11}$$

and consistency with Carroll boosts demands the same scaling for $X_{\mathrm{P}}$.

$$X_{\mathrm{P}} \to \tilde{X}_{\mathrm{P}} = X_{\mathrm{P}} \, e^{-\alpha(X)} \tag{12}$$

The Carroll Casimir equation (8e)

$$\mathrm{d}X_{\mathrm{H}} + \mathcal{V}(X,\, X_{\mathrm{H}}) \, e = 0 \qquad \leftrightarrow \qquad \mathrm{d}\tilde{X}_{\mathrm{H}} + \tilde{\mathcal{V}}(\tilde{X},\, \tilde{X}_{\mathrm{H}}) \, \tilde{e} = 0 \tag{13}$$

establishes the transformation behaviour of the potential

$$\mathcal{V}(X,\, X_{\mathrm{H}}) \to \tilde{\mathcal{V}}(\tilde{X},\, \tilde{X}_{\mathrm{H}}) = e^{-2\alpha(X)} \left( \mathcal{V}(X,\, X_{\mathrm{H}}) - X_{\mathrm{H}}^2 \, \partial_X \alpha \right). \tag{14}$$

The auxiliary field equation (8f) yields an inhomogeneous shift for $\omega$,

$$\omega \to \tilde{\omega} = \omega + (\partial_X \alpha)(X_{\mathrm{H}}\tau + X_{\mathrm{P}}e) \,. \tag{15}$$

The Carroll torsion and curvature equations (8a,8b) are compatible with all the transformations above, replacing consistently everywhere quantities with their tilde counterparts.

Thus, dilaton-dependent Weyl rescalings (9) act pretty much in the same way as in the Lorentzian case (see [55]) and can be used to introduce or eliminate a kinetic potential function $U(X)$ in potentials of the type (16) below.

### 2.1.3   Second-order formulation

To set the stage for other discussions, we translate the first-order/PSM formulation to the second-order formulation.

While the functional dependence of $\mathcal{V}$ can, in principle, be arbitrary, we consider it to be of the form

$$\mathcal{V}(X,\, X_{\mathrm{H}}) = -\frac{U(X)}{2} X_{\mathrm{H}}^2 + V(X) \tag{16}$$

for some kinetic and potential function of the dilaton, $U(X)$ and $V(X)$, respectively. (This is also the most commonly used form of the potential in Lorentzian 2d dilaton gravity [55].)

To get the second-order formulation, one needs to integrate out $\omega$ and $X_{\mathrm{H}}$ by their own equations of motion (8). For this we introduce the dual vectors $v^\mu$ and $e^\mu$ satisfying

$$v^\mu \tau_\mu = -1 \qquad e^\mu e_\mu = 1 \qquad \delta^\mu_\nu = -v^\mu \tau_\nu + e^\mu e_\nu \,. \tag{17}$$

The $\omega$-equation (8d) can be solved algebraically for $X_{\mathrm{H}}$

$$X_{\mathrm{H}} = -e^\mu \partial_\mu X \,, \tag{18}$$

and also leads to a constraint $v^\mu \partial_\mu X = 0$. The $X_{\mathrm{H}}$-equation (8b) is solved by splitting

$$\omega = \hat{\omega} + t + \rho \, e \tag{19}$$

with a torsionless part $\hat{\omega}$ satisfying $d\tau + \hat{\omega} \wedge e = 0$, a torsion part $t$, and an arbitrary undetermined function $\rho$. The latter embodies the usual ambiguity that the Carrollian spin connection is not entirely determined by the equations of motion. Explicitly, the different parts read

$$\hat{\omega}_\mu = -e^\nu \partial_\mu \tau_\nu + e^\nu \partial_\nu \tau_\mu := -2e^\nu \partial_{[\mu} \tau_{\nu]} \qquad\qquad t = U(X)\, X_{\mathrm{H}} \tau \qquad (20)$$

where the latter is determined by the $X_{\mathrm{H}}$-equation. Plugging these solutions into the first-order action (2) with (3) yields

$$I_{1^{\mathrm{st}}} = \frac{k}{2\pi} \int_{\mathcal{M}} \left( X\, d\hat{\omega} - \rho\, dX \wedge e + X_{\mathrm{P}}\, de + \left( -\frac{U(X)}{2} \left(e^\mu \partial_\mu X\right)^2 + V(X) \right) \tau \wedge e \right). \quad (21)$$

It can be seen that $\rho$ plays the role of a Lagrange multiplier for the above-mentioned constraint $v^\mu \partial_\mu X = 0$. The vielbein postulates (see Appendix A.2) relate the connection $\hat{\omega}$ to the Riemann curvature tensor by $R^\lambda{}_{\sigma\mu\nu} = -v^\lambda e_\sigma (d\hat{\omega})_{\mu\nu}$. On the other hand, using the change of basis $dx^\mu = -v^\mu \tau + e^\mu e$, we write $d\hat{\omega} = \partial_{[\mu} \hat{\omega}_{\nu]} dx^\mu \wedge dx^\nu = 2\partial_{[\mu} \hat{\omega}_{\nu]} e^\mu v^\nu \tau \wedge e$ implying $d\hat{\omega} = \frac{R}{2} \tau \wedge e$ where we defined[2] $R = 2e^\mu e^\nu R^\lambda{}_{\mu\lambda\nu}$. Finally, we rewrite $de = 2e^\mu v^\nu \partial_{[\mu} e_{\nu]} \tau \wedge e = K\tau \wedge e$ in terms of the trace of the extrinsic curvature $K$ and define the volume form $\tau \wedge e = \tau_\mu e_\nu\, dx^\mu \wedge dx^\nu = \varepsilon^{\mu\nu} \tau_\mu e_\nu\, d^2x = \det(\tau, e)\, d^2x$.

Inserting these results into the first-order action (21) yields the second-order Carroll dilaton gravity action

$$I_{2^{\mathrm{nd}}}[e, \tau, \rho, X_{\mathrm{P}}, X] = \frac{k}{4\pi} \int_{\mathcal{M}} \mathcal{L}_{2^{\mathrm{nd}}} \qquad (22)$$

with

$$\boxed{\mathcal{L}_{2^{\mathrm{nd}}} = d^2x\, \det(\tau, e) \left( XR + 2\rho\, v^\mu \partial_\mu X + 2X_{\mathrm{P}} K - U(X)\left(e^\mu \partial_\mu X\right)^2 + 2V(X) \right).} \qquad (23)$$

As we will show later, comparing with the literature (e.g. [56–58]) allows identifying this action with the magnetic Carroll theory. This action is Carroll invariant if the Lagrange multipliers transform under Carroll boosts as

$$\delta_\lambda \rho = -U\lambda e^\mu \partial_\mu X + \nabla_\mu (e^\mu \lambda) \qquad\qquad \delta_\lambda X_{\mathrm{P}} = -\lambda e^\mu \partial_\mu X \qquad (24)$$

where $\nabla$ is the connection associated with $\hat{\omega}$ via the vielbein postulates. The left equality is compatible with the transformation behaviour of $\omega$ in (5) by using (19). The right equality agrees with the corresponding on-shell transformation in the first-order formalism (4).

### 2.1.4 PSM formulation

There is another, gauge-theoretic, formulation of 2d Carroll dilaton gravity that lies at the heart of the original construction in [43]. Since some statements are phrased succinctly in this PSM formulation, we briefly summarize its main aspects.

---

[2]This relation between the Riemann tensor and the curvature scalar works in 2d only. In higher dimensions, one can still define a Carroll invariant curvature scalar in terms of Cartan variables (see, e.g., [56]) which, however, cannot be obtained as a contraction of the Riemann tensor. To be explicit, the Carroll invariant curvature scalar in $D > 2$ would read $R = -2v^\mu e_a^\nu R_{\mu\nu}{}^a + e_a^\mu e_b^\nu R_{\mu\nu}^{ab}$, while the Riemann tensor can be expressed (see [41]) as $R^\rho{}_{\sigma\mu\nu} = -v^\rho e_{a\sigma} R_{\mu\nu}{}^a + e_a^\rho e_{\sigma b} R_{\mu\nu}^{ab}$. A naive contraction would give $2e^{a\mu} e_a^\nu R^\lambda{}_{\mu\lambda\nu} = -2v^\mu e_a^\nu R_{\mu\nu}{}^a + 2e_a^\mu e_b^\nu R_{\mu\nu}^{ab}$ which is not Carroll invariant except in 2d. Then, the last term vanishes and $R = 2e^\mu e^\nu R^\lambda{}_{\mu\lambda\nu}$.

PSMs are topological, non-linear gauge theories in 2d [59, 60] and arise as the most general consistent deformation of abelian BF-theories [61]. The PSM bulk action [43]

$$
I_{\mathrm{PSM}}[A_I, X^I] = \frac{k}{2\pi} \int_{\mathcal{M}} \left( X^I \, \mathrm{d}A_I + \frac{1}{2} P^{IJ}(X^K) \, A_I \wedge A_J \right)
\tag{25}
$$

with the field content

$$
A_I = (\omega, \tau, e) \qquad\qquad X^I = (X, X_{\mathrm{H}}, X_{\mathrm{P}})
\tag{26}
$$

and the Poisson tensor

$$
P^{IJ} = \begin{pmatrix} 0 & 0 & X_{\mathrm{H}} \\ 0 & 0 & \mathcal{V}(X, X_{\mathrm{H}}) \\ -X_{\mathrm{H}} & -\mathcal{V}(X, X_{\mathrm{H}}) & 0 \end{pmatrix}
\tag{27}
$$

is equivalent to the first-order action (2). The scalar fields $X^I$ are target space coordinates of a Poisson manifold, and for each of them, there is an associated gauge connection 1-form $A_I$. The non-linear Jacobi identities

$$
P^{IL}\partial_L P^{JK} + P^{JL}\partial_L P^{KI} + P^{KL}\partial_L P^{IJ} = 0
\tag{28}
$$

hold, essentially because the potential depends only on Carroll boost-invariant combination of the target space coordinates $X^I$, viz., $X$ and $X_{\mathrm{H}}$.

The action (25) is invariant under the non-linear gauge symmetries

$$
\delta_\lambda X^J = \lambda_I \, P^{IJ} \qquad\qquad \delta_\lambda A_I = \mathrm{d}\lambda_I + \left(\partial_I P^{JK}\right) A_J \lambda_K \,.
\tag{29}
$$

It is easy to verify that (29) with the choices above is equivalent to (4)-(6).

PSMs have no local physical degrees of freedom. So all physical excitations can be considered holographically as boundary states. Since the Poisson tensor is anti-symmetric and hence must have an even rank, a Poisson tensor associated with a 3-dimensional target space like in (27) necessarily has a non-trivial kernel. Physically, this kernel corresponds to a conserved Casimir that can be interpreted as mass, as we shall see in Section 3.1.

## 2.2   Carroll dilaton gravity as limit

While it is not necessary to do so, quite often it can be helpful to consider Carroll gravity as a singular limit of relativistic gravity, e.g., to build some intuition about the theory and its solution space. Additionally, it can be expedient to think of (some models of) 2d dilaton gravity as coming from the dimensional reduction of higher-dimensional Einstein gravity. We address both issues in this Subsection, beginning with the latter.

To help orient the reader, we provide Fig. 1 which also highlights the consistency, i.e., commutativity, of the various ways we take the limits. In particular, we show that we can first dimensionally reduce and then take the Carroll limit, or the other way around. Readers who do not care about the details of why this is true may take a glance at Fig. 1 and then skip ahead to Subsection 2.3, where we provide all solutions of Carroll 2d dilaton gravity.

### 2.2.1   Dilaton gravity from spherical reduction

For this work to be self-contained, we review the standard spherical reduction of the Einstein–Hilbert action.

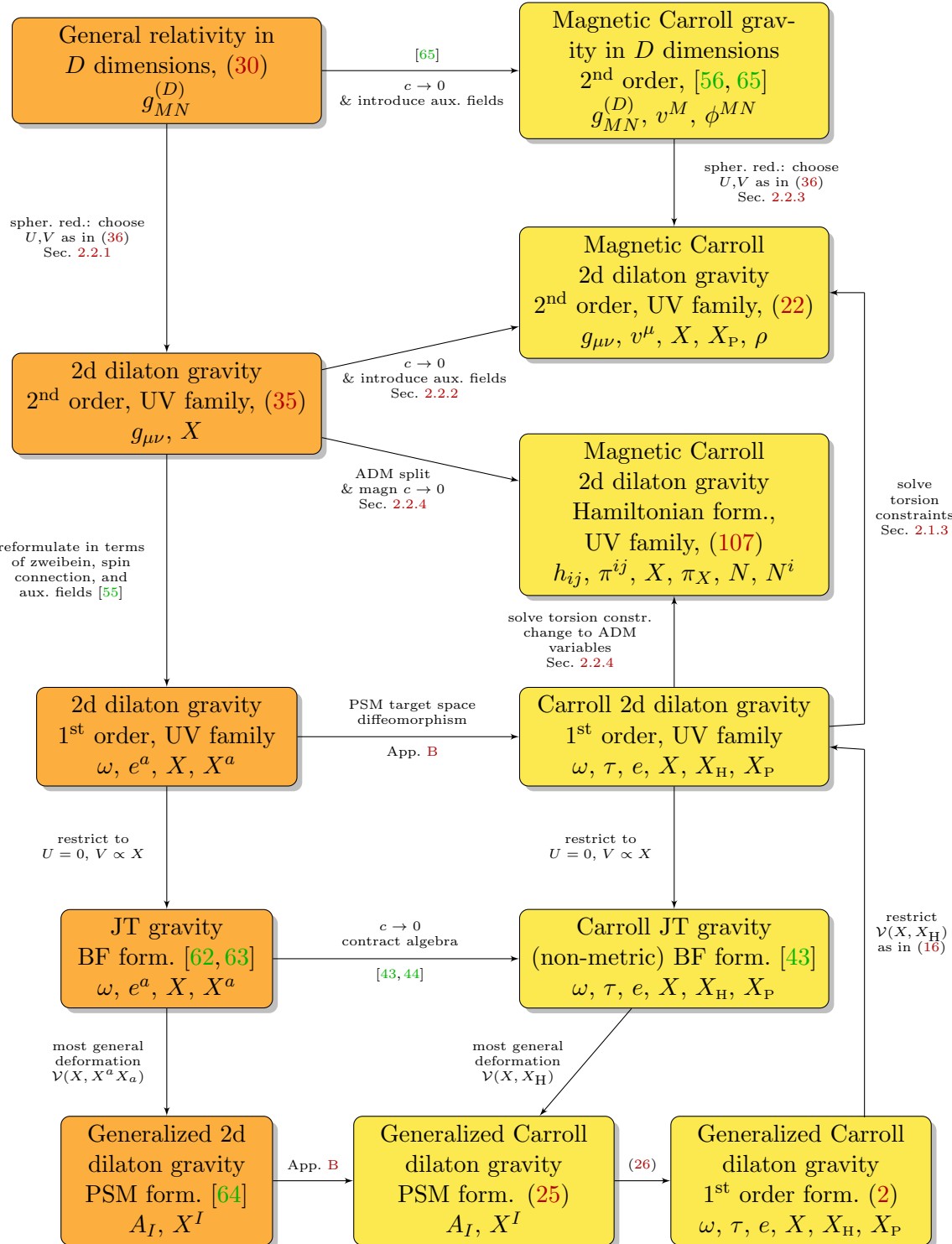

Figure 1: Left (orange): Lorentzian theories. Right (yellow): Carroll theories.

Take Einstein gravity in $D > 3$ spacetime dimensions

$$I_{\text{EH}} = \frac{c^3}{16\pi G} \int \mathrm{d}^D x \sqrt{-g^{(D)}}\, \mathcal{R}^{(D)} \tag{30}$$

and impose spherical symmetry, i.e., assume the isometry group of the metric has $SO(D-1)$ as a (not necessarily proper) subgroup, with $(D-2)$-spheres as orbits. Without loss of generality, pick coordinates adapted to spherical symmetry,

$$\mathrm{d}s^2 = g_{\alpha\beta}(x^\gamma)\, \mathrm{d}x^\alpha\, \mathrm{d}x^\beta + \Phi^2(x^\gamma)\, \mathrm{d}\Omega^2_{S(D-2)} \tag{31}$$

where $\Phi$ is a scalar function depending on the first two coordinates only and $\alpha, \beta, \gamma = 0, 1$. The quantity $\mathrm{d}\Omega^2_{S(D-2)}$ denotes the metric of the round $(D-2)$-sphere. This ansatz splits the higher-dimensional metric into a 2d metric $g_{\alpha\beta}$ and a 2d scalar field $\Phi$, which is precisely the field content of 2d dilaton gravity.

Expressing the Ricci scalar $\mathcal{R}^{(D)}$ in terms of these lower-dimensional quantities yields

$$\mathcal{R}^{(D)} = \mathcal{R} - 2(D-2)\frac{\nabla^2_{(\text{LC})}\Phi}{\Phi} - (D-3)(D-2)\frac{(\partial\Phi)^2}{\Phi^2} + \frac{(D-3)(D-2)}{\Phi^2}\,, \tag{32}$$

where $\mathcal{R}$ and $\nabla_{(\text{LC})}$ are Ricci scalar and covariant derivative associated to the 2d Levi–Civitá connection. The volume form splits into a product of a scalar factor, the 2d volume form, and a volume factor associated with the round $(D-2)$-sphere,

$$\mathrm{d}^D x \sqrt{-g^{(D)}} = \Phi^{D-2}\, \mathrm{d}^2 x \sqrt{-g}\, \mathrm{d}^{D-2} y \sqrt{g^{(D-2)}}\,. \tag{33}$$

To recover 2d dilaton gravity in standard conventions, we redefine

$$\Phi = \frac{D-2}{\lambda} X^{\frac{1}{D-2}} \tag{34}$$

with a constant $\lambda$ of inverse length dimension (to get dimensionless dilaton $X$). After integrating over the $(D-2)$-sphere, and up to total derivatives, the action (30) reduces to

$$I_{\text{sph.red.}}[g_{\alpha\beta}, X] = \frac{k_{\text{rel}}}{4\pi} c^3 \int \mathrm{d}^2 x \sqrt{-g}\, \Big( X\mathcal{R} - U(X)(\partial X)^2 + 2V(X) \Big) \tag{35}$$

with the potentials

$$U(X) = -\frac{D-3}{(D-2)\, X} \qquad\qquad V(X) = \lambda^2\, \frac{D-3}{2(D-2)}\, X^{\frac{D-4}{D-2}} \tag{36}$$

and the coupling constant

$$k_{\text{rel}} = \frac{\pi^{(D-1)/2}}{2\Gamma((D-1)/2)\, G} \left(\frac{D-2}{\lambda}\right)^{(D-2)}\,. \tag{37}$$

In summary, we can describe Schwarzschild–Tangherlini black holes in terms of 2d dilaton gravity, in the sense that the classical solutions of (35) with (36) are in one-to-one correspondence with spherically symmetric solutions to $D$-dimensional Einstein gravity (30). The next step is to take the Carroll limit of the action (35) to obtain Carroll dilaton gravity.

### 2.2.2  Magnetic Carroll dilaton gravity from ultra-relativistic expansion

We start by considering the so-called magnetic limit of Lorentzian 2d dilaton gravity given by the second-order action (35), except that we allow here for arbitrary potentials $U(X)$ and $V(X)$. Using the conventions of [65], we switch to pre-Carrollian variables

$$g_{\mu\nu} = -c^2 T_\mu T_\nu + E_\mu E_\nu \qquad\qquad g^{\mu\nu} = -\frac{1}{c^2} V^\mu V^\nu + E^\mu E^\nu \qquad (38)$$

where

$$V^\mu T_\mu = -1 \qquad\qquad E^\mu E_\mu = 1 \qquad\qquad E^\mu E_\nu - V^\mu T_\nu = \delta^\mu_\nu \qquad (39)$$

and the other contractions are zero. The fields $T_\mu$, $V^\mu$, $E_\mu$, $E^\mu$ and $X$ are assumed to be Taylor-expandable in $c^2$. In particular, we have to leading-order (LO) the Carrollian fields

$$V^\mu = v^\mu + \mathcal{O}(c^2) \qquad\qquad E_\mu = e_\mu + \mathcal{O}(c^2) \qquad\qquad X = X + \mathcal{O}(c^2)\,, \qquad (40)$$

where we denoted the leading-order term in the dilaton expansion by the same letter. As the subleading terms in the $X$-expansion do not play a role in the following and both sides are invariant under Carroll boosts this is just a convenient definition. The first goal is to rewrite all quantities in the relativistic action (35) in terms of the variables on the left-hand side of (40). The relativistic Levi–Civitá connection is thereby organized in a specific way,

$$\overset{\text{(LC)}}{\Gamma}{}^\rho{}_{\mu\nu} = \frac{1}{2} g^{\rho\alpha} \left( \partial_\mu g_{\alpha\nu} + \partial_\nu g_{\alpha\mu} - \partial_\alpha g_{\mu\nu} \right) = \frac{1}{c^2} \overset{(-2)}{C}{}^\rho{}_{\mu\nu} + \tilde{C}^\rho{}_{\mu\nu} + S^\rho{}_{\mu\nu} + c^2 \overset{(2)}{C}{}^\rho{}_{\mu\nu} \qquad (41)$$

where all orders in $c^2$ transform tensorially except the $c^0$ part. This part is further split into a connection $\tilde{C}^\rho{}_{\mu\nu}$ satisfying

$$\overset{(\tilde{C})}{\nabla}{}_\nu E_\mu = 0 \qquad\qquad \overset{(\tilde{C})}{\nabla}{}_\nu V^\mu = 0 \qquad (42)$$

and a tensorial part $S^\rho{}_{\mu\nu}$. In this way, a true Carrollian connection is obtained at leading order in the limit $c \to 0$ (for further details see [65]). While one has to keep in mind that the pre-Carrollian variables $E_\mu$ and $V^\mu$ are still Lorentz-covariant, it is useful to treat them as if they were Carrollian in order to determine the split of the $\mathcal{O}(c^0)$ part of the relativistic Levi–Civitá connection into $\tilde{C}^\rho{}_{\mu\nu}$ and $S^\rho{}_{\mu\nu}$. In other words, we introduce a pre-Carrollian connection[3] $\tilde{\Omega}_\mu$ on the frame bundle which satisfies the Carrollian vielbein postulates

$$\partial_\mu T_\nu - \tilde{C}^\lambda{}_{\mu\nu} T_\lambda + \tilde{\Omega}_\mu E_\nu = 0 \qquad\qquad\qquad \partial_\mu E_\nu - \tilde{C}^\lambda{}_{\mu\nu} E_\lambda = 0 \qquad (43)$$

$$\partial_\mu V^\nu + \tilde{C}^\nu{}_{\mu\lambda} V^\lambda = 0 \qquad\qquad\qquad \partial_\mu E^\nu + \tilde{C}^\nu{}_{\mu\lambda} E^\lambda + V^\nu \tilde{\Omega}_\mu = 0 \qquad (44)$$

where the off-diagonal equations are compatible with (42). Solving for $\tilde{C}^\rho{}_{\mu\nu}$ in terms of $\tilde{\Omega}_\mu$ and the vielbein leads to

$$\tilde{C}^\lambda{}_{\mu\nu} = -V^\lambda \left( \partial_\mu T_\nu + \tilde{\Omega}_\mu E_\nu \right) + \frac{1}{2} \Pi^{\lambda\rho} \left( \partial_\mu \Pi_{\nu\rho} + \partial_\nu \Pi_{\mu\rho} - \partial_\rho \Pi_{\mu\nu} \right) - \Pi^{\lambda\rho} T_\nu \mathcal{K}_{\mu\rho}$$

$$= -V^\lambda \left( \partial_\mu T_\nu + \tilde{\Omega}_\mu E_\nu \right) + E^\lambda \partial_\mu E_\nu \qquad (45)$$

---

[3]In 2d, there is no rotational part of the connection and the boost part has only one component denoted by the one-form $\tilde{\Omega}$.

where $\Pi_{\mu\nu} = E_\mu E_\nu$ and $\mathcal{K}_{\mu\rho}$ is the extrinsic curvature. In 2d, the latter object only has one component,

$$\mathcal{K}_{\mu\nu} = \mathcal{K} E_\mu E_\nu = -\frac{1}{2}\mathcal{L}_V(E_\mu E_\nu) \qquad \Rightarrow \qquad \mathcal{K} = 2E^\mu V^\nu \partial_{[\mu} E_{\nu]} \; . \tag{46}$$

As usual in a second-order formulation, $\tilde{\Omega}_\mu$ is not an independent dynamical variable but is determined in terms of the vielbein such that its torsion vanishes. However, in a (pre-) Carrollian geometry, this can only be achieved to a certain degree as there are torsion components independent of $\tilde{\Omega}_\mu$, corresponding to intrinsic torsion. One, therefore, sets only those torsion components to zero that depend on the pre-Carrollian connection. In 2d, this leads to only one equation,

$$\mathrm{d}T + \tilde{\Omega} \wedge E = 0 \tag{47}$$

solved by

$$\tilde{\Omega}_\mu = -2E^\nu \partial_{[\mu} T_{\nu]} + \gamma(x^\alpha) E_\mu \; . \tag{48}$$

Here, $\gamma$ is some arbitrary function representing the remaining freedom in the pre-Carrollian connection [56,58]. Inserting into (45) yields

$$\tilde{C}^\lambda{}_{\mu\nu} = -V^\lambda \partial_{(\mu} T_{\nu)} - V^\lambda T_{(\mu} \mathcal{L}_V T_{\nu)} + E^\lambda \partial_\mu E_\nu - \gamma V^\lambda E_\mu E_\nu \tag{49}$$

which is not torsion-free, but contains the expected intrinsic torsion $\tilde{C}^\rho{}_{[\mu\nu]} = E^\rho \partial_{[\mu} E_{\nu]}$ even in the limit $c \to 0$. While the arbitrary function $\gamma$ is set to zero in [65], we keep it, for now, to see how it contributes at later stages of the expansion. We will see below that $\gamma$ does not play a role.

If we go back to (41) and use the pre-Carrollian variables we get at order $c^0$

$$\overset{(\mathrm{LC})}{\Gamma}{}^\lambda{}_{\mu\nu}\Big|_{c^0} = -V^\lambda \partial_{(\mu} T_{\nu)} - V^\lambda T_{(\mu} \mathcal{L}_V T_{\nu)} + E^\lambda \partial_\mu E_\nu + E^\lambda E^\alpha T_\nu \mathcal{K}_{\mu\alpha} =: \tilde{C}^\lambda{}_{\mu\nu} + S^\lambda{}_{\mu\nu} \; . \tag{50}$$

Using (49), the expansion of the Levi–Civitá connection (41) has coefficients

$$\overset{(-2)}{C}{}^\rho{}_{\mu\nu} = -V^\rho \mathcal{K}_{\mu\nu} \tag{51}$$

$$\tilde{C}^\rho{}_{\mu\nu} = -V^\rho \partial_{(\mu} T_{\nu)} - V^\rho T_{(\mu} \mathcal{L}_V T_{\nu)} + E^\rho \partial_\mu E_\nu - \gamma V^\rho E_\mu E_\nu \tag{52}$$

$$S^\rho{}_{\mu\nu} = E^\rho E^\lambda T_\nu \mathcal{K}_{\mu\lambda} + \gamma V^\rho E_\mu E_\nu \tag{53}$$

$$\overset{(2)}{C}{}^\rho{}_{\mu\nu} = -E^\rho E^\alpha T_{(\mu}(\mathrm{d}T)_{\nu)\alpha} \; . \tag{54}$$

The Riemann tensor associated with the Levi–Civitá connection is defined by

$$\mathcal{R}^\lambda{}_{\mu\nu\sigma} = \partial_\nu \overset{(\mathrm{LC})}{\Gamma}{}^\lambda{}_{\sigma\mu} + \overset{(\mathrm{LC})}{\Gamma}{}^\lambda{}_{\nu\beta} \overset{(\mathrm{LC})}{\Gamma}{}^\beta{}_{\sigma\mu} - (\nu \leftrightarrow \sigma) \; . \tag{55}$$

We work directly with the Ricci tensor $\mathcal{R}_{\mu\nu} := \mathcal{R}^\lambda{}_{\mu\lambda\nu}$ in the following. It can be organized as

$$\mathcal{R}_{\mu\nu} = \frac{1}{c^4}\overset{(-4)}{R}{}_{\mu\nu} + \frac{1}{c^2}\overset{(-2)}{R}{}_{\mu\nu} + \overset{(0)}{R}{}_{\mu\nu} + c^2 \overset{(2)}{R}{}_{\mu\nu} + c^4 \overset{(4)}{R}{}_{\mu\nu} \tag{56}$$

with the coefficients up to $\mathcal{O}(c^2)$ given by

$$\overset{(-4)}{R}{}_{\mu\nu} = 0 \tag{57}$$

$$\overset{(-2)}{R}{}_{\mu\nu} = \overset{(\tilde{C})}{\nabla}_\lambda \overset{(-2)}{C}{}^\lambda{}_{\nu\mu} - 2\tilde{C}^\alpha{}_{[\nu\mu]}\overset{(-2)}{C}{}^\lambda{}_{\alpha\mu} + S^\lambda{}_{\lambda\beta}\overset{(-2)}{C}{}^\beta{}_{\nu\mu} - S^\alpha{}_{\nu\lambda}\overset{(-2)}{C}{}^\lambda{}_{\alpha\mu} \tag{58}$$

$$\overset{(0)}{R}{}_{\mu\nu} = \overset{(\tilde{C})}{R}{}_{\mu\nu} + \overset{(\tilde{C})}{\nabla}_\lambda S^\lambda{}_{\nu\mu} - \overset{(\tilde{C})}{\nabla}_\nu S^\lambda{}_{\lambda\mu} - 2\tilde{C}^\lambda{}_{[\nu\beta]}S^\beta{}_{\lambda\mu} - \overset{(-2)}{C}{}^\lambda{}_{\nu\beta}\overset{(2)}{C}{}^\beta{}_{\lambda\mu} - \overset{(2)}{C}{}^\lambda{}_{\nu\beta}\overset{(-2)}{C}{}^\beta{}_{\lambda\mu} \tag{59}$$

$$\overset{(2)}{R}{}_{\mu\nu} = \overset{(\tilde{C})}{\nabla}_\lambda \overset{(2)}{C}{}^\lambda{}_{\nu\mu} - 2\tilde{C}^\alpha{}_{[\nu\lambda]}\overset{(2)}{C}{}^\lambda{}_{\alpha\mu} - \overset{(2)}{C}{}^\lambda{}_{\nu\beta}S^\beta{}_{\lambda\mu} - \overset{(2)}{C}{}^\lambda{}_{\alpha\mu}S^\alpha{}_{\nu\lambda} \; . \tag{60}$$

We use these expressions to compute the Ricci scalar expansion

$$\mathcal{R} = -\frac{1}{c^4} V^\mu V^\nu \overset{(-2)}{R}_{\mu\nu} + \frac{1}{c^2}\left( E^\mu E^\nu \overset{(-2)}{R}_{\mu\nu} - V^\mu V^\nu \overset{(0)}{R}_{\mu\nu} \right)$$
$$- V^\mu V^\nu \overset{(2)}{R}_{\mu\nu} + E^\mu E^\nu \overset{(0)}{R}_{\mu\nu} + \mathcal{O}(c^2) \qquad (61)$$

leading to

$$\overset{(-4)}{R} = 0 \qquad (62)$$

$$\overset{(-2)}{R} = -\overset{(\tilde{C})}{\nabla}_\mu\left( V^\mu \mathcal{K} \right) + \mathcal{K}^2 \qquad (63)$$

$$\overset{(0)}{R} = E^\mu E^\nu \overset{(\tilde{C})}{R}_{\mu\nu} - \overset{(\tilde{C})}{\nabla}_\rho\left( E^\rho \overset{(\tilde{C})}{\nabla}_\mu E^\mu \right) + \overset{(\tilde{C})}{\nabla}_\rho\left( V^\rho \gamma \right) - \mathcal{K}\gamma \qquad (64)$$

$$= E^\mu E^\nu \overset{(\tilde{C})}{R}_{\mu\nu}\Big|_{\gamma=0} - \overset{(\tilde{C})}{\nabla}_\rho\left( E^\rho \overset{(\tilde{C})}{\nabla}_\mu E^\mu \right) . \qquad (65)$$

In the last equality, we used that the last two terms in (64) cancel with the $\gamma$-dependence in the first term such that all the $\gamma$-dependence in the last line is in the total derivative term. In the following, we use that the Carroll compatible derivative allows writing a total divergence as

$$\int \mathrm{d}^2x \, \det(T,E) \, \overset{(\tilde{C})}{\nabla}_\mu X^\mu = -\int \mathrm{d}^2x \, \det(T,E) \, \mathcal{K} T_\mu X^\mu \qquad (66)$$

up to boundary terms [65]. We are ready to expand in $c^2$ and rewrite the relativistic dilaton gravity action (35) in terms of the pre-Carrollian variables.

$$I_{\mathrm{dil}} = c^2 \overset{(2)}{I} + c^4 \overset{(4)}{I} + \mathcal{O}(c^6) \qquad (67)$$

The first two terms in this expansion are

$$\overset{(2)}{I} = \frac{k_{\mathrm{rel}}}{4\pi} \int \mathrm{d}^2x \, \det(T,E) \left( \mathcal{K} V^\mu \partial_\mu X + U(V^\mu \partial_\mu X)^2 \right) \qquad (68)$$

$$\overset{(4)}{I} = \frac{k_{\mathrm{rel}}}{4\pi} \int \mathrm{d}^2x \, \det(T,E) \left( X E^\mu E^\nu \overset{(\tilde{C})}{R}_{\mu\nu}\Big|_{\gamma=0} - U(X)(E^\mu \partial_\mu X)^2 + 2V(X) \right.$$
$$\left. - X \overset{(\tilde{C})}{\nabla}_\rho\left( E^\rho \overset{(\tilde{C})}{\nabla}_\mu E^\mu \right) \right) . \qquad (69)$$

To extract the magnetic action, we rewrite $\overset{(2)}{I}$ as

$$\overset{(2)}{I} = \frac{k_{\mathrm{rel}}}{4\pi} \int \mathrm{d}^2x \, \det(T,E) \left( \frac{1}{2}\mathcal{K} V^\mu \partial_\mu X + \frac{1}{2}\left( \mathcal{K} + 2U V^\mu \partial_\mu X \right) V^\mu \partial_\mu X \right)$$
$$= \frac{k_{\mathrm{rel}}}{4\pi} \int \mathrm{d}^2x \, \det(T,E) \left( -c^4 \frac{2\mathcal{K}}{V^\mu \partial_\mu X} X_{\mathrm{P}}^2 + c^2 2 X_{\mathrm{P}} \mathcal{K} \right.$$
$$\left. - c^4 \frac{2V^\mu \partial_\mu X}{\mathcal{K} + 2U V^\mu \partial_\mu X} \rho^2 + c^2 2\rho V^\mu \partial_\mu X \right) \qquad (70)$$

where on-shell evaluation of the auxiliary fields $X_{\mathrm{P}}$ and $\rho$ reproduces the action in the first line. The introduction of these auxiliary fields effectively redistributes the powers of $c$ such that $\overset{(2)}{I}$ is converted into terms contributing to $\overset{(4)}{I}$ and $\overset{(6)}{I}$. The magnetic theory is obtained by rescaling Newton's constant as $G \to c^4 G_M$, defining $k := k_{\mathrm{rel}} c^4$ and taking the limit $c \to 0$. Effectively, this picks out the action $\overset{(4)}{I}$ together with the auxiliary

field contributions from $\overset{(2)}{I}$ at leading order. In this case the connection $\tilde{C}$ reduces to the Carroll compatible connection $\Gamma$ satisfying $\nabla(e_\mu e_\nu) = 0 = \nabla v^\mu$, where $\nabla$ is the associated derivative. The dust settles and we obtain the magnetic Carroll dilaton gravity action

$$I_{\text{mag}}^L[e, \tau, \rho, X_{\text{P}}, X] = \frac{k}{4\pi} \int \mathcal{L}_{\text{mag}}^L \tag{71}$$

with

$$\mathcal{L}_{\text{mag}}^L = \mathrm{d}^2 x \, \det(\tau, e) \Big( X(e^\mu e^\nu R_{\mu\nu}\big|_{\gamma=0} - \nabla_\mu a^\mu) + 2\rho \, v^\mu \partial_\mu X + 2X_{\text{P}} K$$
$$- U(X)(e^\mu \partial_\mu X)^2 + 2V(X) \Big) \tag{72}$$

where it was assumed that the potential $V$ does not scale with $c$ and the leading term of $\mathcal{K}$ is denoted by $K$. The field $X_{\text{P}}$ acts as a Lagrange multiplier setting the intrinsic torsion of $\Gamma^\rho{}_{\mu\nu}$ given by the extrinsic curvature $K$ to zero. On-shell this leaves us with a Carrollian torsion-free connection satisfying the requirements of [58]. The spatial acceleration vector $a^\mu$ is defined by

$$a^\mu = e^\mu e^\nu a_\nu \qquad a_\mu = 2v^\nu \partial_{[\nu} \tau_{\mu]} \tag{73}$$

as in [66]. As divergence terms $\nabla_\mu X^\mu$ are independent of the ambiguous $\gamma$-dependent term in the Carroll compatible connection, we see that $\gamma$ does not enter in this action. Therefore we could have set it to zero from the beginning, without loss of generality, like in [65]. The curvature term can be further massaged by using the identity

$$e^\mu e^\nu R_{\mu\nu}\big|_{\gamma=0} = -e^\mu e^\nu \nabla_\mu a_\nu - (e^\mu a_\mu)^2 = -\nabla_\mu a^\mu \tag{74}$$

which holds only in 2d. Using additionally the definition of the Carrollian curvature scalar $R = 2e^\mu e^\nu R_{\mu\nu}\big|_{\gamma=0}$ we find that the Lagrange-2-form (72) matches with (23).

### 2.2.3 Spherical reduction of magnetic Carroll gravity

Let us look at the other corner of Fig. 1 and see if the two paths from $D$-dimensional Einstein gravity to 2d magnetic Carroll dilaton gravity commute. For this, we start with the magnetic limit of $D$-dimensional Einstein gravity, impose spherical symmetry, and reduce the corresponding action by the angular variables. We approach this in the second-order formulation, i.e., we work with the spin connection expressed in terms of the vielbein variables by using the torsion constraints. Nevertheless, we will not insert the expressions explicitly to keep the notation cleaner. The $D$-dimensional torsion components read

$$T^0 = \mathrm{d}\tau + \omega^a \wedge e_a \qquad T^a = \mathrm{d}e^a + \omega^{ab} \wedge e_b \tag{75}$$

where $a = 1, ..., D-1$ and $\omega^{ab} = -\omega^{ba}$. By spherical symmetry, we write the Carroll metric as

$$h_{MN} \, \mathrm{d}x^M \, \mathrm{d}x^N = h_{\mu\nu}(x^\sigma) \, \mathrm{d}x^\mu \, \mathrm{d}x^\nu + \Phi^2(x^\sigma) \, \mathrm{d}\Omega_{S^{(D-2)}}^2 \; . \tag{76}$$

It is convenient to phrase things in terms of a choice of vielbein,

$$h = e \otimes e + \delta_{lm} e^l \otimes e^m = \bar{e} \otimes \bar{e} + \Phi^2 \delta_{lm} \bar{e}^l \otimes \bar{e}^m \tag{77}$$

where the internal indices take values $l, m = 2, ..., D-1$ and in the second equality we defined a transverse vielbein $\bar{e}^l$ normalized to the unit sphere $S^{(D-2)}$. To keep the notation simple we set $e^1 \equiv e$, $\omega^1 \equiv \omega$. The relations between the vielbeins are

$$\bar{\tau}_M = \tau_M \qquad \bar{e}_M = e_M \qquad \bar{e}_M^l = \Phi^{-1} e_M^l \tag{78}$$
$$\bar{v}^M = v^M \qquad \bar{e}^M = e^M \qquad \bar{e}_l^M = \Phi \, e_l^M \tag{79}$$

where $\tau_M v^M = -1$, $e_M e^M = 1$, and $e^l_M e^M_m = \delta^l_m$ define the dual versions. We assume throughout that the coordinates are chosen in such a way that only $\bar{e}^l_M$ depend on the internal coordinates. The barred vielbeins are assumed to satisfy the Carrollian torsion constraints, which as a reminder, are not obtained by setting all the torsion to zero but only those components that allow the elimination of the spin connection from the first-order action. We have

$$\mathrm{d}\bar{\tau} + \bar{\omega} \wedge \bar{e} = 0 \qquad\qquad \mathrm{d}\bar{e}^l + \bar{\omega}^{lm} \wedge \bar{e}_m = 0 \tag{80}$$

where the remaining torsion component $\mathrm{d}\bar{e}$ crucially is not set to zero. The second of these equations corresponds to the spherical part which is thus fixed to be torsion-free. Let us assume $\bar{\omega} = \omega$ and $\bar{\omega}^{lm} = \omega^{lm}$. Furthermore, we impose vanishing torsion in the unbarred space by the same principle. In particular, this means that the following conditions are imposed

$$T^0(e_a, v) = T_a(e_b, e_c) = T^0(e_a, e_b) = T_{[a}(e_{b]}, v) = 0 \; . \tag{81}$$

This sets to zero all the higher-dimensional torsion except the components $T_{(a}(e_{b)}, v)$ which correspond to the intrinsic torsion (see, e.g., Appendix B of [65]). The conditions can be solved for $\omega^{1l} = -\omega^{l1}$ and $\omega^l$ by

$$\omega^l = \varphi \, \bar{e}^l \qquad\qquad \omega^{l1} = (e^\mu \partial_\mu \Phi)\bar{e}^l \tag{82}$$

where $\varphi$ is an arbitrary function. This can be inserted to compute the decomposition of the scalar curvature

$$R^{(D)} = -2v^M e^N_a \Omega^a_{MN} + e^M_a e^N_b \Omega^{ab}_{MN} \tag{83}$$

where

$$\Omega^a = \mathrm{d}\omega^a + \omega^{ab} \wedge \omega_b \qquad\qquad \Omega^{ab} = \mathrm{d}\omega^{ab} + \omega^a{}_c \wedge \omega^{cb} \; . \tag{84}$$

Explicitly, the components are given by

$$\Omega^1 = \bar{\Omega}^1 = \mathrm{d}\bar{\omega} \qquad\qquad \Omega^{1l} = -\mathrm{d}(e^\mu \partial_\mu \Phi) \wedge \bar{e}^l = -\Omega^{l1} \tag{85}$$

$$\Omega^l = \mathrm{d}\varphi \wedge \bar{e}^l + (e^\mu \partial_\mu \Phi)\bar{e}^l \wedge \bar{\omega} \qquad\qquad \Omega^{lm} = \bar{\Omega}^{lm} - (e^\mu \partial_\mu \Phi)\bar{e}^l \wedge \bar{e}^m \tag{86}$$

leading to the scalar curvature

$$R^{(D)} = -4v^\mu e^\nu \partial_{[\mu}\bar{\omega}_{\nu]} + \frac{2(D-2)}{\Phi}\Big((e^\mu\partial_\mu\Phi)v^\nu\bar{\omega}_\nu - v^\mu\partial_\mu\varphi - e^\mu\partial_\mu(e^\alpha\partial_\alpha\Phi)\Big)$$
$$- (D-2)(D-3)\frac{(e^\mu\partial_\mu\Phi)^2}{\Phi^2} + \frac{(D-2)(D-3)}{\Phi^2} \tag{87}$$

where we used $R_{S^{(D-2)}} = (D-2)(D-3)$. Plugging this result into the $D$-dimensional action (see [58]), using the field redefinition (34), and integrating over the angles yields

$$I_{\mathrm{Car}} = \frac{1}{16\pi G_M} \int \mathrm{d}^D x \det(\tau, e^a) \, R^{(D)} \tag{88}$$

$$= \frac{k}{4\pi} \int \mathrm{d}^2 x \det(\tau, e)\Big(4X e^\mu v^\nu \partial_{[\mu}\bar{\omega}_{\nu]} - 2\lambda\varphi X^{\frac{3-D}{2-D}} K + 2\lambda\varphi\frac{D-3}{D-2}X^{\frac{1}{2-D}}v^\mu\partial_\mu X$$
$$+ 2v^\mu\bar{\omega}_\mu e^\nu\partial_\nu X - 2e^\mu\partial_\mu(e^\alpha\partial_\alpha X) - U(e^\mu\partial_\mu X)^2 + 2V\Big) \tag{89}$$

with $k$, $U$, and $V$ defined in the same way as in Section 2.2.1 but $G$ exchanged with the magnetic version $G_M$. The curvature terms simplify by noticing that

$$-2e^\mu\partial_\mu(e^\alpha\partial_\alpha X) = -2X\nabla_\mu a^\mu \qquad\qquad 2v^\mu\bar{\omega}_\mu e^\nu\partial_\nu X = 2X\nabla_\mu a^\mu \tag{90}$$

up to total derivatives. These two terms, therefore, cancel in the action. The quantity $\bar{\omega}$ is the 2d spin connection evaluated on a solution of the torsion constraints, i.e.,

$$\bar{\omega} = \hat{\omega} + \rho e \tag{91}$$

with an undetermined component given by the arbitrary function $\rho(x^\alpha)$. Together with the definition of the curvature scalar $R = 4e^\mu v^\nu \partial_{[\mu}\hat{\omega}_{\nu]}$ the action reads

$$I_{\text{Car}} = \frac{k}{4\pi} \int \mathrm{d}^2 x \det(\tau, e) \Big( X R + 2\rho v^\mu \partial_\mu X + 2X_{\text{P}} K - U(e^\mu \partial_\mu X)^2 + 2V \Big) \tag{92}$$

where we discarded all boundary terms and redefined the Lagrange multipliers $\rho$, $X_{\text{P}}$ as specific combinations of the free functions $\rho$ and $\varphi$,

$$\rho \to \rho - \varphi\lambda\frac{D-3}{D-2}X^{\frac{1}{2-D}} \qquad\qquad \varphi \to -\frac{X_{\text{P}}}{\lambda}X^{\frac{3-D}{D-2}} \ . \tag{93}$$

This matches with the result (23) and shows that spherical reduction and taking the magnetic limit commute. A similar result was found already for the Galilean case [67].

### 2.2.4 Magnetic Carroll dilaton gravity from a Hamiltonian perspective

Let us introduce ADM variables [68] and perform the computation that coined the term "magnetic" for 2d dilaton gravity. For a comparison with the Einstein-gravity case, see, e.g., [58]. Introducing a foliation $\Sigma_t$ with a time function $t$ we choose adapted coordinates $(t, x^i)$ and write the metric as

$$\mathrm{d}s^2 = -c^2 N^2 \, \mathrm{d}t^2 + h_{ij}\big( \mathrm{d}x^i + N^i \, \mathrm{d}t \big)\big( \mathrm{d}x^j + N^j \, \mathrm{d}t \big) \tag{94}$$

such that we get a future-directed timelike unit normal vector

$$n = \frac{1}{Nc}\big(\partial_t - N^i \partial_i\big) \qquad\qquad n^2 = -1 \ . \tag{95}$$

The label $x^i$ here actually only refers to a single coordinate since we are in 2d. In these adapted coordinates, the contracted Gauss–Codazzi equation reads

$$R = R^{(1)} + K^2 + K^{ij} K_{ij} - \frac{2}{Nc}\big(\partial_t K - N^i \partial_i K\big) - \frac{2}{N} D_i D^i N \tag{96}$$

where $D_i$ is the Levi–Civitá derivative associated to $h_{ij}$. We use $K_{ij} = K h_{ij}$ and $R^{(1)} = 0$ in 2d. Furthermore, the trace of extrinsic curvature can be written as

$$K = -\frac{1}{2Nc} h^{ij}\big(\dot{h}_{ij} - 2D_i N_j\big) \tag{97}$$

where the dot denotes a partial derivative with respect to $t$. This allows writing the relativistic second-order action (35) in terms of ADM variables,

$$I_{\text{dil}}[X, h_{ij}, N, N^i] = \int \mathrm{d}t \, L \tag{98}$$

with

$$L = \frac{k_{\text{rel}}c^3}{4\pi} \int \mathrm{d}x \sqrt{h} \Big( 2K\big(\dot{X} - N^i \partial_i X\big) - 2c\, N h^{ij} D_i D_j X$$
$$+ \frac{U}{Nc}\big(\dot{X} - N^i \partial_i X\big)^2 - Nc\, U h^{ij} D_i X D_j X + 2Nc V \Big) \ . \tag{99}$$

Defining momentum densities

$$\pi_X = \frac{\delta L}{\delta \dot{X}} = \frac{k_{\rm rel} c^3 \sqrt{h}}{4\pi} \left(2K + 2U\left(\dot{X} - N^i \partial_i X\right)\right) \tag{100}$$

$$\pi^{ij} = \frac{\delta L}{\delta \dot{h}_{ij}} = -\frac{k_{\rm rel} c^2 \sqrt{h}}{4\pi N} h^{ij}\left(\dot{X} - N^i \partial_i X\right) \tag{101}$$

we perform a Legendre transformation and write the action in Hamiltonian form

$$I_{\rm dil}[X, \pi_X, h_{ij}, \pi^{ij}] = \int {\rm d}t\,{\rm d}x \left(\pi_X \dot{X} + \pi^{ij} \dot{h}_{ij} - N\mathcal{H} - N^i \mathcal{H}_i\right) \tag{102}$$

where the momentum constraint is

$$\mathcal{H}_i = \pi_X D_i X - 2D_j \pi^j{}_i \tag{103}$$

and the Hamiltonian constraint reads

$$\mathcal{H} = \mathcal{H}^M + \mathcal{H}^E \tag{104}$$

$$\mathcal{H}^M = \frac{k\sqrt{h}}{4\pi}\left(U h^{ij} D_i X D_j X - 2V + 2h^{ij} D_i D_j X\right) \tag{105}$$

$$\mathcal{H}^E = -\frac{4\pi c^2}{k\sqrt{h}}\left(\pi^{ij} h_{ij} \pi_X + U\pi^{ij} \pi_{ij}\right) . \tag{106}$$

In the above expression, we already rescaled $G \to G_M c^4$ and defined $k := k_{\rm rel} c^4$ [cf. (37)] such that the leading order in a $c \to 0$ expansion corresponds to the magnetic dilaton gravity action

$$\boxed{I_{\rm mag}^H[X, \pi_X, h_{ij}, \pi^{ij}] = \int {\rm d}t\,{\rm d}x \left(\pi_X \dot{X} + \pi^{ij} \dot{h}_{ij} - N\mathcal{H}^M - N^i \mathcal{H}_i\right) .} \tag{107}$$

As in the case of Einstein gravity, the momenta cannot be eliminated by their equations of motion anymore as they only appear linearly. Instead, they act as Lagrange multipliers for the (second-class) constraints

$$\delta\pi_X : \qquad \dot{X} - N^i \partial_i X = 0 \qquad \leftrightarrow \qquad n^\mu \partial_\mu X = 0 \tag{108}$$

$$\delta\pi^{ij} : \qquad \dot{h}_{ij} - 2D_{(i} N_{j)} = 0 \qquad \leftrightarrow \qquad K_{ij} = 0 \tag{109}$$

which are the same constraints obtained in the first-order approach if we identify $v^\mu \propto n^\mu$ and $h_{ij} = e_i e_j$. Let us see if the actions themselves are also equivalent. The action (23) reads

$$I_{\rm 2nd} = \frac{k}{2\pi}\int {\rm d}^2 x\,\det(\tau, e)\left(2X e^\mu v^\nu \partial_{[\mu} \hat{\omega}_{\nu]} + \rho v^\mu \partial_\mu X + X_{\rm P} K - \frac{U}{2}(e^\mu \partial_\mu X)^2 + V\right) . \tag{110}$$

We have to identify these variables in terms of the ADM variables used before in adapted coordinates $(t, x)$. Let us first fix the boost frame such that $\tau = N\,{\rm d}t$. The frame variables then read

$$v^\mu = \left(-\frac{1}{N}, \frac{\mathcal{N}}{N}\right) \qquad \tau_\mu = \left(N, 0\right) \qquad e^\mu = \left(0, \frac{1}{\mathfrak{e}}\right) \qquad e_\mu = \left(\mathcal{N}\mathfrak{e}, \mathfrak{e}\right) . \tag{111}$$

This allows writing the torsionless part of the spin connection as

$$\hat{\omega}_t = 2e^\mu \partial_{[t} \tau_{\mu]} = -\frac{\partial_x N}{\mathfrak{e}} \qquad\qquad \hat{\omega}_x = 0 \tag{112}$$

while we also have the relations

$$K = 2e^\mu v^\nu \partial_{[\mu} e_{\nu]} = \frac{1}{2N\mathfrak{e}}\left(\partial_t \mathfrak{e}^2 - 2D_x(\mathcal{N}\mathfrak{e}^2)\right) \qquad \det(\tau, e) = N\mathfrak{e} \ . \tag{113}$$

Note that $\mathcal{N}$ are the components of a spatial vector and $\mathfrak{e}^2\mathcal{N}$ the components of a spatial covector such that the spatial covariant derivative in the first expression reads explicitly

$$D_x(\mathcal{N}\mathfrak{e}^2) = \partial_x(\mathcal{N}\mathfrak{e}^2) - \gamma(\mathcal{N}\mathfrak{e}^2) = \mathfrak{e}\partial_x(\mathcal{N}\mathfrak{e}) \tag{114}$$

where the 1d Christoffel symbols are $\gamma = \partial_x \ln \mathfrak{e}$. This allows writing the first term in the action as

$$2X e^\mu v^\nu \partial_{[\mu}\hat{\omega}_{\nu]} = -\frac{X}{N\mathfrak{e}^2}D_x^2 N \ . \tag{115}$$

In total, we arrive at

$$I_{2\mathrm{nd}} = \frac{k}{2\pi}\int \mathrm{d}t\,\mathrm{d}x\Big(-\frac{X}{\mathfrak{e}}D_x^2 N - \mathfrak{e}\rho(\partial_t X - \mathcal{N}D_x X) \tag{116}$$

$$+\frac{X_\mathrm{P}}{2\mathfrak{e}}\left(\partial_t\mathfrak{e}^2 - 2D_x(\mathcal{N}\mathfrak{e}^2)\right) - \frac{N}{2\mathfrak{e}}U(D_x X)^2 + VN\mathfrak{e}\Big) \tag{117}$$

which, by the change of variables

$$-\frac{k}{2\pi}\mathfrak{e}\rho = \pi_X \qquad\qquad \frac{k}{4\pi}\frac{X_\mathrm{P}}{\mathfrak{e}} = \pi_h \tag{118}$$

can be brought into the form

$$I_{2\mathrm{nd}}[X, \pi_X, \mathfrak{e}, \pi_h, N, \mathcal{N}] = \int \mathrm{d}t\, L_{2\mathrm{nd}} \tag{119}$$

with

$$L_{2\mathrm{nd}} = \int \mathrm{d}x\Big(\pi_X \dot{X} + \pi_h \partial_t \mathfrak{e}^2 - \mathcal{N}\big(\pi_X D_x X - 2\mathfrak{e}^2 D_x \pi_h\big) \tag{120}$$

$$- N\mathfrak{e}\frac{k}{2\pi}\Big(\frac{1}{\mathfrak{e}^2}D_x^2 X + \frac{U}{2\mathfrak{e}^2}(D_x X)^2 - V\Big)\Big) \ . \tag{121}$$

Changing back to spatial component notation

$$\mathfrak{e} \to e_i \qquad \mathfrak{e}^{-1} \to e^i \qquad \pi_h \to \pi^{ij} \qquad D_x \to D_i \qquad \mathcal{N} \to N^i \tag{122}$$

yields

$$L_{2\mathrm{nd}} = \int \mathrm{d}x\Big(\pi_X \dot{X} + \pi^{ij}\partial_t h_{ij} - N^i\big(\pi_X D_i X - 2D_j \pi^j{}_i\big) \tag{123}$$

$$- N\mathfrak{e}\frac{k}{4\pi}\Big(2h^{ij}D_i D_j X + U h^{ij}D_i X D_j X - 2V\Big)\Big) \ . \tag{124}$$

We thus find that the second-order formulation and the Hamiltonian formulation agree.

$$I_{2\mathrm{nd}} = I_{\mathrm{mag}}^H \tag{125}$$

## 2.3 Solutions of Carroll dilaton gravity

In order to derive all classical solutions, the first-order/PSM formulation is advantageous. In the end, we translate our solutions to the second-order formulation.

### 2.3.1    Constant dilaton vacua

Constant dilaton vacua are defined to be solutions where $X_\mathrm{H}$ vanishes everywhere. The equation that normally determines the Carroll metric (8d) leads to constant dilaton (hence the name). The Carroll Casimir equation (8e) shows that this constant cannot be anything but has to solve the non-differential equation

$$\mathcal{V}(X, 0) = 0\,. \tag{126}$$

In particular, this means constant dilaton vacua need infinite finetuning of the dilaton field and may not even exist for some models (the simplest example is the Carroll CGHS model where $\mathcal{V} = \Lambda \neq 0$).

The Carroll curvature of all constant dilaton vacua is a constant times the volume-form,

$$\Omega = -\partial_X \mathcal{V}(X, 0)\,\tau \wedge e\,. \tag{127}$$

Similar remarks apply to torsion. Finally, the auxiliary field equation (8f) implies that also $X_\mathrm{P}$ is some (arbitrary) constant.

Since constant dilaton vacua are non-generic, require infinite finetuning, and are not very rich in structure, we move on to the generic sector, the linear dilaton vacua.

### 2.3.2    Linear dilaton vacua

Linear dilaton vacua are solutions where $X_\mathrm{H}$ does not vanish everywhere. Thus, let us start by assuming a patch where $X_\mathrm{H} \neq 0$.[4] This allows to solve the Carroll metric equation (8d) as

$$e = -\frac{\mathrm{d}X}{X_\mathrm{H}}\,. \tag{128}$$

Inserting this result into the Carrollian Casimir equation (8e) yields

$$\frac{1}{2}\,\mathrm{d}(X_\mathrm{H}^2) - \mathcal{V}(X,\,X_\mathrm{H})\,\mathrm{d}X = 0 \tag{129}$$

which allows expressing $X_\mathrm{H}$ as function of the dilaton $X$ and of an integration constant $M$. We refer to $M$ as Carrollian mass or Carrollian Casimir. The latter nomenclature was chosen since in the PSM formulation, the function $M(X, X_\mathrm{H})$ spans precisely the kernel of the degenerate Poisson tensor (27). I.e., if we went to Casimir–Darboux coordinates the expression $M(X, X_\mathrm{H})$ would correspond to the Casimir direction in the Poisson manifold.

To simplify the discussion, we assume, for now, $\mathcal{V} = V(X)$ and return to more general cases in the end. It is useful to define the integrated potential

$$w(X) := \int^X V(y)\,\mathrm{d}y \tag{130}$$

in terms of which the conserved Carrollian mass is given by

$$M = w(X) - \frac{1}{2}\,X_\mathrm{H}^2 \qquad\qquad \mathrm{d}M = 0\,. \tag{131}$$

The equation that establishes no intrinsic torsion, (8c), is solved trivially,

$$\mathrm{d}e = 0 \qquad \Longrightarrow \qquad e = \mathrm{d}r\,. \tag{132}$$

---

[4]Below, many statements implicitly come with the qualifier "assuming $X_\mathrm{H} \neq 0$".

We use $r$ as our Carrollian radial coordinate without loss of generality.[5] Expressing $X_{\mathrm{H}}$ as a function of $X$ using the Carrollian mass (131) and inserting it into (128) yields a simple differential equation for the dilaton

$$\frac{\mathrm{d}X}{\mp\sqrt{2(w(X) - M)}} = \mathrm{d}r \qquad (133)$$

where the signs refer to the two branches of the square-root function.

To solve the remaining equations, it is convenient to fix the Carroll boosts such that $X_{\mathrm{P}} = 0$, which is always possible locally. The auxiliary equation (8f) simplifies to a constraint

$$\omega = -\frac{V}{X_{\mathrm{H}}}\tau \qquad (134)$$

that renders the remaining two equations, for Carroll curvature (8a) and torsion (8b), identical to each other.

Thus, there is only one more equation we need to solve, e.g., the Carrollian torsion equation (8b). By virtue of the constraint (134) it simplifies to

$$\mathrm{d}\tau + \big(\partial_X \ln X_{\mathrm{H}}\big)\tau \wedge \mathrm{d}X = 0 \qquad (135)$$

solved by

$$\tau = -X_{\mathrm{H}}\,\mathrm{d}t\,. \qquad (136)$$

Here, we used the scaling ambiguity $t \to \alpha\tilde{t}$ with $\alpha \in \mathbb{R}^+$ to choose some time coordinate $t$ and fixed the residual Carroll boost invariance by assuming $\tau$ has no $\mathrm{d}r$-component. The result (136) implies

$$\omega = V(X)\,\mathrm{d}t \qquad (137)$$

for the Carroll boost connection. As a consequence, the Carrollian curvature of our solutions is given by

$$\Omega = -V'(X)\,\tau \wedge e = -V'(X)\,\mathrm{d}t \wedge \mathrm{d}X\,. \qquad (138)$$

In summary, in the chosen gauge, the solution reads

$$X = \text{given by integrating (133)} \qquad \omega = V(X)\,\mathrm{d}t \qquad (139a)$$
$$X_{\mathrm{H}} = \pm\sqrt{2(w(X) - M)} \qquad\qquad \tau = -X_{\mathrm{H}}\,\mathrm{d}t \qquad (139b)$$
$$X_{\mathrm{P}} = 0 \qquad\qquad\qquad\qquad e = \mathrm{d}r\,. \qquad (139c)$$

Translating our solution to the second-order formulation as described in Section 2.1.3 yields the metric

$$\mathrm{d}s^2 = h_{\mu\nu}\,\mathrm{d}x^\mu\,\mathrm{d}x^\nu = e_\mu e_\nu\,\mathrm{d}x^\mu\,\mathrm{d}x^\nu = \mathrm{d}r^2 \qquad (140)$$

and the timelike vector field

$$v = v^\mu \partial_\mu = \frac{1}{X_{\mathrm{H}}}\,\partial_t = \pm\frac{1}{\sqrt{2(w(X) - M)}}\,\partial_t\,. \qquad (141)$$

---

[5]This is true only if we disregard edge modes, which we do for the time being. Once a boundary is considered with specific boundary conditions imposed on the fields, there could be a loss of generality in assuming $e = \mathrm{d}r$.

The dilaton field $X$ is still given by integrating (133).

Finally, we come back to more general cases when the potential $\mathcal{V}$ does not only depend on the dilaton $X$ but additionally depends on the boost invariant scalar $X_{\mathrm{H}}$. We discuss here the family of models (16) and refer by analogy to [64] for further generalizations.

We continue to fix the radial coordinate by $e = \mathrm{d}r$, so the Carroll metric is still given by (140). Exploiting the Weyl rescalings discussed after (9), we find that we need a modified definition of the function $w$, namely

$$w(X) := \int^X e^{Q(y)} V(y) \, \mathrm{d}y \qquad\qquad e^{Q(X)} := e^{\int^X U(y) \, \mathrm{d}y} \,. \tag{142}$$

The additive integration constant implicit in the function $w(X)$ can be absorbed by shifting the mass $M$. The multiplicative integration constant implicit in $e^{Q(X)}$ can be chosen to give this expression the desired physical dimensions, discussed in Subsection 3.4 below.

Relatedly, the Carrollian mass is changed slightly

$$M = w(X) - \frac{1}{2} X_{\mathrm{H}}^2 \, e^{Q(X)} \tag{143}$$

which changes also the boost invariant scalar

$$X_{\mathrm{H}} = \pm \sqrt{2 e^{-Q(X)} (w(X) - M)} \,. \tag{144}$$

The dilaton is obtained by integrating

$$\frac{\mathrm{d}X}{\mp \sqrt{2 e^{-Q(X)} (w(X) - M)}} = \mathrm{d}r \,. \tag{145}$$

Fixing, again, Carroll boosts suitably recovers $X_{\mathrm{P}} = 0$ and

$$\tau = -e^{Q(X)} \, X_{\mathrm{H}} \, \mathrm{d}t \,. \tag{146}$$

The analogue of the constraint (134) implies

$$\omega = e^{Q(X)} \, \mathcal{V}(X, \, X_{\mathrm{H}}) \, \mathrm{d}t \,. \tag{147}$$

Finally, the timelike vector field is given by

$$v = v^\mu \partial_\mu = \pm \frac{1}{\sqrt{2 e^{Q(X)} (w(X) - M)}} \, \partial_t \,. \tag{148}$$

Carroll curvature evaluates as

$$\Omega = -\big(V'(X) - \tfrac{1}{2} X_{\mathrm{H}}^2 U'(X)\big) \tau \wedge e = e^{Q(X)} \big(\tfrac{1}{2} X_{\mathrm{H}}^2 U'(X) - V'(X)\big) \, \mathrm{d}t \wedge \mathrm{d}X \,. \tag{149}$$

### 2.3.3 Carrollian Birkhoff theorem

In Lorentzian dilaton gravity, there is a generalized Birkhoff theorem, in the sense that all solutions to all models have at least one Killing vector, see e.g. [55] and references therein.

In the Carrollian case, we see similar features: in the constant dilaton sector, all solutions have constant curvature and constant scalar fields, so there is a sense in which these configurations are maximally symmetric. However, let us focus on the less trivial linear dilaton vacua.

A minimal requirement to define a Carrollian Killing vector is that all boost-invariant fields are invariant under Lie transport along it. This establishes the conditions

$$\mathcal{L}_\xi h_{\mu\nu} = \mathcal{L}_\xi v^\mu = \mathcal{L}_\xi X = 0 \,. \tag{150}$$

The last condition implies $\xi^r = 0$, the second condition yields $\partial_t \xi^t = 0$, and the first condition imposes no further restriction. Thus, any vector field

$$\xi^\mu \, \partial_\mu = f(r) \, \partial_t \tag{151}$$

is a Carrollian Killing vector (including $\xi^\mu = v^\mu$), for all solutions of all 2d Carroll dilaton gravity models. We refer to this statement as "Carrollian Birkhoff theorem". Since all Carrollian Killing vectors (151) mutually commute, one can refer to them as radial supertranslations by analogy to BMS jargon.

### 2.3.4   Singularities of Carrollian manifolds

It is not the purpose of our work to provide a comprehensive discussion of Carrollian singularities. Nevertheless, we need to confront three types of singularities since they naturally and rather generically occur in 2d Carroll dilaton gravity.

1. Carroll coordinate singularities

2. Carroll curvature singularities

3. Carrollian structure singularities

The first type of coordinate singularity can arise much in the same way as in general relativity. The prototypical example is Schwarzschild-gauge, which in our context is obtained by using the radial coordinate

$$\mathrm{d}\rho = e^{Q(X)} \, \mathrm{d}X \tag{152}$$

in terms of which the Carroll metric reads

$$h_{\mu\nu} \, \mathrm{d}x^\mu \, \mathrm{d}x^\nu = \frac{\mathrm{d}\rho^2}{2e^{Q(X)}(w(X) - M)} \,. \tag{153}$$

There is a (coordinate-)singularity at zeros of $w(X) - M$. We shall use this singular coordinate system when comparing with Schwarzschild–Tangherlini solutions in Section 5.

Concerning curvature singularities, we merely note that they can (and quite often do) arise, typically in the strong-coupling region where the dilaton tends to zero. Perhaps there is nothing more to this type of singularity than the observation that curvature can become infinite if gravity is infinitely strong.

The third type is a singularity reminiscent of what happens at the bifurcation sphere of the Schwarzschild black hole (see, e.g., [46]), in the following sense. The attentive reader will have realized already that there is a remaining issue in our classification of solutions: first, we assumed $X_{\mathrm{H}} = 0$ everywhere (constant dilaton vacua) and then we assumed $X_{\mathrm{H}} \neq 0$ everywhere (linear dilaton vacua) but what if $X_{\mathrm{H}}$ vanishes or diverges at isolated loci? That this can happen is evident from the explicit solution (139) for linear dilaton vacua: The equation $X_{\mathrm{H}} = 0$ can have solutions for certain values of the radial coordinate $r$, depending on the choice of the potential and the value of the mass $M$. Thus, we have potentially singular points in the interior or the boundary of linear dilaton vacua. The difference to the Lorentzian case is that there the singularity analogous to the one at $X_{\mathrm{H}} = 0$ is merely a coordinate singularity of Eddington–Finkelstein patches and can be removed by going into, say, Kruskal coordinates, see e.g. [69]. By contrast, in the Carrollian case, there is no coordinate system where both the metric and the vector field are non-zero and finite at $X_{\mathrm{H}} = 0$.

The physical interpretation of these potential singularities is simple: the timelike vector field either blows up ($X_{\mathrm{H}} \to 0$) or collapses to zero ($|X_{\mathrm{H}}| \to \infty$). We call this a singularity in the Carrollian structure. Note that curvature (138) may remain finite at such loci.

In Section 4, where we define Carroll extremal surfaces, we employ these intriguing loci where $X_{\mathrm{H}}$ vanishes, independently from the behaviour of curvature.

$$X_{\mathrm{H}} = 0 \qquad \leftrightarrow \qquad e^{-Q(X)}(w(X) - M) = 0 \tag{154}$$

We move on to the global properties of Carroll thermal manifolds next, where such loci also play a decisive role.

### 2.3.5   Global aspects of Carroll thermal solutions

Before we provide further motivation and details, let us define a Carroll thermal (C-thermal) manifold:

**C-thermal manifolds** (in 2d) are smooth manifolds $\mathcal{M}$ carrying a Carrollian structure up to isolated points, with a boundary $\partial\mathcal{M}$ diffeomorphic to $S^1$.

Thus, we are relaxing the original definition of a Carrollian manifold [4], which disallowed these isolated singularities in the Carrollian structure. As we do not investigate any form of matter coupled to this theory, the word "thermal" has to be understood in a broader sense than being able to read off an actual temperature from some detector. We shall see in Section 3 that it is still natural to use this terminology because of a corresponding term appearing in the first law. To amalgamate these conflicting indicators in favour and against a Carroll version of temperature, we refer to these geometries as C-thermal.

In analogy to the Euclidean case, we compactify the orbits of a Carrollian Killing vector field $\xi$ associated with time translations. Taking $\xi = \partial_t$ we identify[6] points $p \in \mathcal{M}$ along the action

$$p \sim e^{\beta\xi} \cdot p \qquad\qquad \beta \in \mathbb{R}^+ \tag{155}$$

where $e^{\beta\xi} \cdot$ means flowing the point $p$ along the integral curve of $\xi$ by a parameter difference $\beta$.

Having in mind a Carrollian analogue of Euclidean dilaton gravity, we introduce a (cut-off) boundary at some large positive value of the dilaton, $X(r_c) = X_c$. The two topologies of interest to us are cylinder and disk. C-thermal manifolds, by definition, require the latter.

The natural topology for a 2d Carrollian manifold with compactified time direction is a cylinder, i.e., a direct product manifold of the spatial line and the temporal cycle. Assuming the temporal cycle to be finite and non-zero globally, only cylinders are possible. Thus, if we insisted on the absence of Carrollian structure singularities C-thermal manifolds, which have disk topology, would be impossible.

A smooth disk is obtained by demanding the temporal cycle to shrink to a point at the locus $X_{\mathrm{H}} \to 0$ such that the manifold is smooth there. Before tackling this issue, we address general aspects of Carroll manifolds with a Carrollian structure singularity induced by $X_{\mathrm{H}} \to 0$ in the centre of the disk.

The frame we define on such a manifold is given by

$$v = \frac{1}{e^Q X_{\mathrm{H}}} \, \partial_t \qquad\qquad e = \partial_r \tag{156}$$

---

[6]As in the Lorentzian case, we analytically continue to imaginary time and change to complexified frame variables and spin connection, see Section 6 for an explicit example. Unlike the Lorentzian case, this has no consequence on the signature, which remains $(0, +)$. To reduce clutter we refrain from introducing Wick rotated variables here.

where the coordinate $t$ is compactified, $t \sim t + \beta$. Similarly to a frame adapted to polar coordinates on a Euclidean disk, it becomes singular at the origin. This is not only because of the divergence in $v$ but also because the very notion of tangent and radial directions cannot be defined at the origin. The way to still obtain a global orthonormal frame field in the Euclidean case is to perform an $SO(2)$-rotation to a Cartesian frame that can be extended to the origin. In the general case, this could be done locally such that, e.g., asymptotically one still has a polar frame while one switches to a Cartesian one in a neighbourhood of the centre with corresponding transition functions on the overlap.

In the Carrollian case, however, this is not possible: The transformations acting on the frame belong to the homogeneous Carroll group $\mathrm{Carr}(2, \mathbb{R}) = ISO(1)$, which leaves $v$ invariant. So, starting with (156) asymptotically, one necessarily arrives at a singular description of the origin. This is just another way of stating the presence of a Carrollian structure singularity at this locus.

One can quantify this singularity by picking a loop around the origin $\gamma : [0, 1] \rightarrow \mathcal{M}$ parametrized by $\sigma \mapsto x^\mu(\sigma)$ and computing its associated holonomy $H_\gamma(\omega)$ for the connection (147). The parallel transport equation

$$\frac{\mathrm{d}}{\mathrm{d}\sigma} V^a + \frac{\mathrm{d}x^\mu}{\mathrm{d}\sigma} \omega^a{}_{\mu b} V^b = 0 \tag{157}$$

is solved by

$$V^a\big(\gamma(\sigma = 1)\big) = \exp\left[ -\int_0^1 \mathrm{d}\sigma \, \dot{x}^\mu \omega_\mu \right]^a{}_b V^b\big(\gamma(\sigma = 0)\big) = \big(H_\gamma\big)^a{}_b V^b\big(\gamma(\sigma = 0)\big) \tag{158}$$

where $\dot{x}^\mu$ denotes $\frac{\mathrm{d}}{\mathrm{d}\sigma} x^\mu$ and $\dot{x}^\mu \omega_\mu$ is a matrix with components $\dot{x}^\mu \omega^a{}_{\mu b}$. Using the solution (147) and choosing the loop $x^\mu(\sigma) = \big(\sigma\beta, r_0\big)$ with $r_0 = \mathrm{const.}$ we obtain

$$H_\gamma(\omega) = \begin{pmatrix} 1 & -\beta e^Q \mathcal{V}(X, X_\mathrm{H})\big|_{r=r_0} \\ 0 & 1 \end{pmatrix} . \tag{159}$$

Then, contracting $\gamma$ to a point at the origin yields

$$\lim_{r_0 \to 0} H_\gamma(\omega) = \begin{pmatrix} 1 & -\beta w'(X) \\ 0 & 1 \end{pmatrix} \bigg|_{r=0} \tag{160}$$

where, without loss of generality, the integration constant in (145) was chosen such that $X_\mathrm{H}(r = 0) = 0$. We stress that while the Carroll spin connection is ambiguous and only defined up to the addition of a term $\rho\, e$ (see Section 2.1.3), this ambiguity does not enter here because the loop is chosen such that $\dot{x}^\mu e_\mu = 0$.

Let us return to the smoothness condition at the origin of the disk. In a Euclidean theory of gravity, one requires that the closer $\gamma$ approaches the origin, the more the holonomy approaches the one of a flat disk. The geometry of the latter is given by

$$v^{\mathrm{E,Disk}} = \frac{1}{r} \partial_\varphi \qquad e^{\mathrm{E,Disk}} = \partial_r \qquad \big(\omega^{\mathrm{E,Disk}}\big)^a{}_b = \begin{pmatrix} 0 & \mathrm{d}\varphi \\ -\mathrm{d}\varphi & 0 \end{pmatrix} \tag{161}$$

with the torsion-free spin connection $\omega^{\mathrm{E,Disk}}$ and $\varphi \sim \varphi + 2\pi$. Explicitly, the condition reads

$$\lim_{r_0 \to 0} H_\gamma(\omega^{\mathrm{E}}) \stackrel{!}{=} H_\gamma(\omega^{\mathrm{E,Disk}}) = \exp\begin{pmatrix} 0 & -2\pi \\ 2\pi & 0 \end{pmatrix} \tag{162}$$

where $\omega^E$ is some curved connection. This condition fixes the Hawking temperature. We use the same definition for the Carrollian case: A flat Carrollian disk is given by

$$v^{\text{C,Disk}} = \frac{1}{r} \partial_\varphi \qquad e^{\text{C,Disk}} = \partial_r \qquad \left(\omega^{\text{C,Disk}}\right)^a{}_b = \begin{pmatrix} 0 & \text{d}\varphi \\ 0 & 0 \end{pmatrix} , \qquad (163)$$

where the ambiguity in the spin connection has again been neglected because it does not contribute to the holonomy integral. The condition we arrive at is

$$\lim_{r_0 \to 0} H_\gamma(\omega) \overset{!}{=} H_\gamma(\omega^{\text{C,Disk}}) = \exp \begin{pmatrix} 0 & -2\pi \\ 0 & 0 \end{pmatrix} \qquad (164)$$

Therefore, in Carrollian theories of gravity, the holonomy is never equal to the identity for contractible loops around the origin, which is precisely because of the Carrollian structure singularity.

As another equivalent way to ensure a smooth disk, we use the Gauss–Bonnet formula rewritten in first-order variables,

$$2\pi\chi = \int_\mathcal{M} \text{d}\omega - \int_{\partial\mathcal{M}} \omega \qquad (165)$$

where implicitly, we assume the bulk term $\text{d}\omega$ does not yield $\delta$-like contributions (corresponding to deficit angles).[7] This assumption, in general, is incorrect unless the periodicity $\beta$ takes certain values.

In other words, while often a formula like (165) is used to compute $\chi$ for given geometrical data on a manifold, we reverse the logic: Taking $\chi = 1$ and demanding

$$2\pi \overset{!}{=} \int_\mathcal{M} \text{d}\omega - \int_{\partial\mathcal{M}} \omega \qquad (166)$$

ensures that no conical defects appear while $\mathcal{M}$ is topologically a disk provided the periodicity $\beta$ is chosen appropriately, which we shall do in the next Section. Here, the boundary integral is understood along the surface $X = X_c$ with outward-pointing unit normal form $n = -X_{\text{H}}^{-1} \text{d}X$ and with a volume form $\text{vol}_{\partial\mathcal{M}}$ induced by $\tau \wedge e = n \wedge \text{vol}_{\partial\mathcal{M}}$ such that Stokes' theorem holds. In particular, this implies $\int_{\partial\mathcal{M}} \omega = -\int_0^\beta e^Q \mathcal{V} \, \text{d}t$.

While the spin connection is undefined at the origin of the disk, the integrand of (166) is defined up to this isolated point, and we can continuously extend $\text{d}\omega = 2e^\mu v^\nu \partial_{[\mu}\omega_{\nu]}\tau \wedge e$ to the origin. On-shell the limit

$$\lim_{r \to 0} 2e^\mu v^\nu \partial_{[\mu}\omega_{\nu]}\big|_{\text{EOM}} = -w''(X)\big|_{r=0} \qquad (167)$$

exists whenever $w''(X)|_{r=0}$ is finite. Thus, we can continue the integrand to the origin in such cases. As the resulting contribution to the integral has measure zero, we find that the formula (166) is not even sensitive to the Carrollian structure singularity and therefore provides another suitable device to probe disk topology. We shall see in the next Section how (166) can be used to fix the Carrollian temperature in terms of the dilaton potential.

---

[7]In general, there is an additional subtlety. Namely, depending on the frame, one may need to subtract an auxiliary connection $\omega_0$ to ensure gauge invariance. The boundary integral is equivalent to an integral of the second fundamental form. However, in our chosen frame, this turns out to be unnecessary.

# 3 Carroll thermal properties

In this Section, we discuss the C-thermal properties of the linear dilaton solutions derived in Section 2.3.2. In Subsection 3.1, we derive the energy from the usual boundary charges. In Subsection 3.2, we define two different notions of temperature and show that they coincide with each other. In Subsection 3.3, we address entropy and the first law of Carrollian thermodynamics. In Subsection 3.4, we perform a dimensional analysis to ensure dimensionless entropy. Finally, in Subsection 3.5, we calculate the specific heat.

## 3.1 Energy

The canonical codimension-2 charge variations for a generic PSM (25) are (see e.g. Eq. (6.1) in [70])

$$\oint \delta Q_\lambda = \frac{k}{2\pi} \lambda_I \, \delta X^I \Big|_{\partial \mathcal{M}} \tag{168}$$

with boundary condition-preserving gauge parameters[8] $\lambda_I$. We assume that the boundary conditions imposed on $X$ and $X_{\mathrm{H}}$ are such that they allow arbitrary variations of the mass parameter $M$. Since we do not want to be too specific about these boundary conditions at this stage, we just impose that diffeomorphisms generated by the Killing vector $\xi = \partial_t$ are part of the asymptotic symmetries that preserve the boundary conditions. The associated gauge parameters are given by $\lambda_X = \omega_t = e^Q \mathcal{V}$, $\lambda_H = \tau_t = -e^Q X_{\mathrm{H}}$, $\lambda_P = e_t = 0$.

The charge variation (168) associated with unit time translations is given by

$$\delta Q_{\partial_t} = \frac{k}{2\pi} \left( e^Q \mathcal{V}(X, X_{\mathrm{H}}) \, \delta X - e^Q X_{\mathrm{H}} \, \delta X_{\mathrm{H}} \right)\big|_{\partial \mathcal{M}} = \frac{k}{2\pi} \delta M \tag{169}$$

where in the last equality, we used the (variation of the) Casimir relation (131). Since $M$ is totally conserved, $\mathrm{d}M = 0$, it does not matter where this quantity is evaluated, which is why we dropped the indicator $|_{\partial \mathcal{M}}$.

The charge (169) is integrable in field space and gives a simple expression for the energy, $E = Q_{\partial_t}$, in terms of the mass parameter:

$$\boxed{E = \frac{k}{2\pi} M} \tag{170}$$

## 3.2 Temperature

Following the discussion in Section 2.3.5 we impose equation (166) to ensure having a Carrollian disk without any defects. Inserting the solutions of Section 2.3.2 and choosing an orientation such that $\tau \wedge e =: e^Q \, \mathrm{d}t \, \mathrm{d}X$ we find

$$\int_{\mathcal{M}} \mathrm{d}\omega = -\int_{\mathcal{M}} \partial_X \mathcal{V} \, \tau \wedge e = \int_0^\beta \int_{X_{\min}}^{X_c} \mathrm{d}t \, \mathrm{d}X \, \partial_X \Big( U(w - M) - \partial_X w \Big) \tag{171}$$

$$= \beta \Big( U(w - M) - \partial_X w \Big)\Big|_{X_{\min}}^{X_c} \tag{172}$$

$$\int_{\partial \mathcal{M}} \omega = -\int_0^\beta e^Q \mathcal{V} \, \mathrm{d}t \Big|^{X_c} = -\beta \Big( \partial_X w - U(w - M) \Big)\Big|^{X_c} \tag{173}$$

---

[8]These parameters, in general, depend on the fields, which can make this expression non-integrable in field space. To account for this possibility, we denote the charge variation by $\oint \delta$.

such that (166) reads

$$2\pi \overset{!}{=} \beta \, \partial_X w \big|_{X_{\min}} \, . \tag{174}$$

Here $X_{\min}$ is the value of the dilaton at the locus $X_{\mathrm{H}} = 0$, taking the positive branch in (133). Interpreting $\beta = T^{-1}$ as inverse Carrollian temperature establishes

$$\boxed{T = \frac{w'(X_{\min})}{2\pi} \, .} \tag{175}$$

The result for the Carrollian temperature (175) is equivalent to the corresponding Lorentzian result for the Hawking temperature of 2d dilaton gravity with the same potentials (16).

In addition to this topological derivation of Carrollian temperature, there is also a definition in terms of Carrollian surface gravity.

$$\nabla_\mu \big(e^Q e^\nu \partial_\nu X\big)\big|_{X_{\mathrm{H}}=0} =: \kappa \, e_\mu \big|_{X_{\mathrm{H}}=0} \tag{176}$$

The quantity in parentheses is proportional to $X_{\mathrm{H}}$ on-shell and thus vanishes at $X_{\mathrm{H}} = 0$. In this sense, the definition of $\kappa$ in (176) is analogous to the definition of surface gravity in a Lorentzian theory. Taking the solutions (139) yields $\kappa = w'(X_{\min})$. Therefore, we recover the anticipated relation

$$T = \frac{\kappa}{2\pi} \tag{177}$$

between Carrollian temperature $T$ and Carrollian surface gravity $\kappa$.

## 3.3 Entropy and first law

As the last missing piece for the first law, let us inspect the definition of the entropy along the lines of Wald [71]. Working in the covariant phase space formalism of first-order Carroll dilaton gravity (see, e.g., [72]) the symplectic form is given by

$$\omega(\delta_1 \phi, \delta_2 \phi) = \frac{k}{2\pi} \Big( \delta_2 X^I \delta_1 A_I - \delta_1 X^I \delta_2 A_I \Big) \tag{178}$$

where we used PSM variables (26) for convenience and denote the collection of fields by $\phi$. Contracting in a diffeomorphism generated by some vector field $\xi$ on the worldsheet and evaluating on-shell yields the fundamental theorem of covariant phase space

$$\omega(\delta\phi, \delta_\xi \phi) \approx \mathrm{d}\Big(\delta Q_\xi - Q_{\delta\xi} - i_\xi \Theta(\delta\phi)\Big) =: \mathrm{d}\slashed{Q}_\xi \tag{179}$$

where $Q_\xi$ is the Noether–Wald charge. The variation of the codimension-2 charges is given by

$$\slashed{Q}_\xi = \frac{k}{2\pi} \xi^\mu A_{I\,\mu} \, \delta X^I \, , \tag{180}$$

which just reproduces the special case $\lambda_I = A_{I\mu}\xi^\mu$ of the more general result (168). We choose $\xi$ to be the Carrollian Killing vector associated with unit time translations,

$$\xi = \partial_t \tag{181}$$

which implies $\omega(\delta\phi, \delta_\xi \phi) = 0$. Additionally, we pick a constant time hypersurface $\Sigma$ extending from the point $\mathcal{E} = \{p \in \mathcal{M} : X_{\mathrm{H}} = 0\}$ in the interior to a point $\mathcal{B} \in \partial\mathcal{M}$ on the asymptotic boundary such that $\partial\Sigma = \mathcal{E} \cup \mathcal{B}$. Integrating (179) over $\Sigma$ leads to the on-shell identity

$$\int_\Sigma \omega(\delta\phi, \delta_\xi \phi) \approx \int_\Sigma \mathrm{d}\slashed{Q}_\xi = \slashed{Q}_\xi \big|_{\mathcal{B}} - \slashed{Q}_\xi \big|_{\mathcal{E}} = 0 \, . \tag{182}$$

Explicitly, $\oint \mathcal{Q}_\xi$ reads on-shell

$$\oint \mathcal{Q}_\xi = \frac{k}{2\pi}\Big(e^Q\Big(V - \frac{U}{2}X_{\mathrm{H}}^2\Big)\delta X - e^Q X_{\mathrm{H}}\delta X_{\mathrm{H}}\Big) \tag{183}$$

and evaluates at the two points of $\partial\Sigma$ to

$$\oint \mathcal{Q}_\xi\Big|_{\mathcal{B}} = \frac{k}{2\pi}\,\delta M \qquad\qquad \oint \mathcal{Q}_\xi\Big|_{\mathcal{E}} = \frac{k}{2\pi}e^Q V\,\delta X\Big|_{\mathcal{E}}\,. \tag{184}$$

From (182) we therefore find

$$\frac{k}{2\pi}\,\delta M = \Big(\frac{e^{Q(X)}V(X)}{2\pi}\,k\,\delta X\Big)\Big|_{\mathcal{E}} \tag{185}$$

which together with the results (170) and (175) takes the form of a first law,

$$\delta E = T\,\delta S\,. \tag{186}$$

The new thermodynamic quantity defined here is the entropy

$$\boxed{\; S := k\,X_{\min}\,. \;} \tag{187}$$

and arises as a Noether charge, just as in the relativistic case. Its functional form in terms of the dilaton also matches precisely with the one of relativistic dilaton gravity [73]. In words, entropy is given by the value of the dilaton at the Carroll extremal surface (defined in the next Section), times the coupling constant.

### 3.4 A word about dimensions

If we assign standard units to all variables in the action, then entropy turns out to have a non-standard dimension of velocity. The quickest way to see this is first to verify that the connection $\omega$ necessarily has the dimension of inverse velocity, assuming that $\tau$ has time dimension and $e$ length dimension. This statement follows from the Carroll torsion equation (8b). The first term in the action (2) has a dimension of $[kX\omega]$ and, in units where $\hbar = 1$, this combination must be dimensionless. Thus, we find that the dimension of entropy (187), $[S] = [kX] = \frac{\text{length}}{\text{time}} = \text{velocity}$, is unusual.

If one wants to recover a dimensionless entropy (e.g. measured in $e$-bits), one has to introduce a velocity as a conversion factor. In Carrollian theories, there is no natural choice for such a conversion factor, but if viewed as an expansion of a Lorentzian theory, we have a velocity available, namely the speed of light. Even for intrinsic Carrollian theories, we shall assume the presence of some quantity with the dimension of velocity and convert time into length. This assumption permits $\tau$ to have length dimension and hence $\omega$ to be dimensionless. In the following, we denote the length dimensions by integers $[\bullet] = n$, meaning that the corresponding quantity $\bullet$ has length dimension $n$ (if $n$ is negative, the quantity $\bullet$ has corresponding inverse length dimension). For instance, our choice above means $[e] = [\tau] = [r] = [t] = 1$ and $[\omega] = 0$.

The remaining freedom is which length dimension to assign to the dilaton field. From an intrinsic 2d perspective, the only natural choice is to assume dimensionless dilaton, $[X] = 0$, implying also $[k] = 0$. The dimensions of all other quantities follow from this assignment and compatibility with the equations of motion (8): $[\mathcal{V}] = -2$, $[X_{\mathrm{H}}] = [X_{\mathrm{P}}] = [M] = [w] = [v] = -1$, $[\Omega] = 0$, $[e^Q] = +1$. The only subtlety (known already from the Lorentzian counterpart) is the last assignment and can be attributed to a length dimension

carried by the (otherwise irrelevant) multiplicative integration constant inherent to the definition of $e^Q$, see (142).[9]

We deduce the dimensions of our thermodynamical quantities as $[E] = [T] = -1$ and $[S] = 0$. In particular, the entropy is dimensionless, and inverse temperature $\beta$ has length dimension consistently with our starting point of assigning time a length dimension.

Our conclusions of this Section are in line with previous results in the literature: standard thermal partition functions in Carroll theories are not well-defined [13, 74], and Carroll quantum field theories suffer from infinite degeneracies [13, 74, 75] that persist for finite volumes [74], so applying the usual rules of statistical mechanics to simple Carroll systems (such as free quantum fields and gases of free particles) does not lead to sensible results. This is not to say that there is no notion of Carroll thermodynamics but the rules of the game are yet to be spelt out. Our notion of thermodynamics for Carroll black holes developed in this Section, including the dimensional analysis above, are steps in this direction.

## 3.5 Specific heat

With the quantities obtained so far, we define a Carrollian specific heat as

$$C := \frac{\mathrm{d}E}{\mathrm{d}T} = T \frac{\mathrm{d}S}{\mathrm{d}T} \tag{188}$$

yielding

$$C = k \frac{w'(X_{\mathrm{min.}})}{w''(X_{\mathrm{min.}})} \ . \tag{189}$$

The equivalence of this expression to the Lorentzian case [76] is worth highlighting. Assuming positive temperature, $w'(X_{\mathrm{min.}}) > 0$, specific heat is positive if and only if $w''(X_{\mathrm{min.}}) > 0$.

Having investigated the C-thermal properties of linear dilaton solutions with Carrollian structure singularities, we define Carroll extremal surfaces and Carroll black holes in the next Section.

# 4 Carroll extremal surfaces

In this Section, we introduce the geometric notion of Carroll extremal surfaces, guided by corresponding Lorentzian results. We start in Subsection 4.1 with a translation of standard (relativistic) extremal surfaces into the PSM formulation. We copy this definition in Subsection 4.2 and apply it to define Carroll extremal surfaces. We translate back this definition into first- and second-order formulations of Carroll gravity in Subsection 4.3. Finally, we collect our results to define Carroll black holes in Subsection 4.4.

## 4.1 Standard extremal surfaces in PSM formulation

Our first task is to translate the notion of an extremal surface (both null expansions vanish, see, e.g. [46]) into the PSM formulation. For relativistic 2d dilaton gravity in the PSM formulation the Poisson tensor is given by [64]

$$P^{IJ} = \begin{pmatrix} 0 & -X^+ & X^- \\ P^{+\mathrm{x}} = -P^{\mathrm{x}+} & 0 & \mathcal{V}(X, X^+ X^-) \\ P^{-\mathrm{x}} = -P^{\mathrm{x}-} & P^{-+} = -P^{+-} & 0 \end{pmatrix} \tag{190}$$

---

[9]Since $[X] = 0$ but $[\mathcal{V}] = -2$, the potential generically contains some relevant coupling constant. We can always use an appropriate power of that constant for the multiplicative integration constant in $e^Q$.

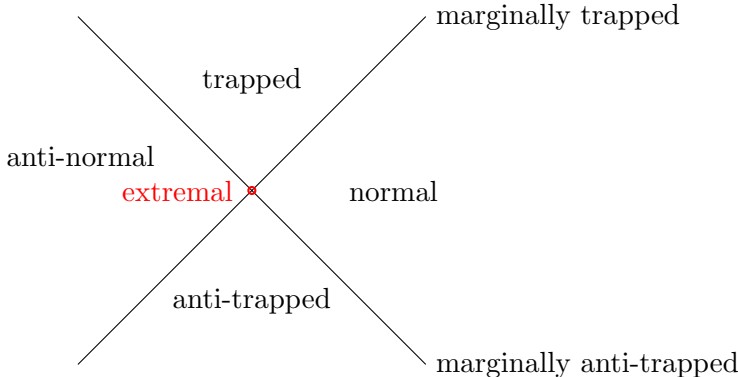

Figure 2: Kruskal diagram of Lorentzian eternal black hole

and the worldsheet metric is written in terms of the zweibein as

$$\mathrm{d}s^2 = 2e^+ e^- \ . \tag{191}$$

On-shell the latter is given by

$$\mathrm{d}s^2 = 2e^Q \, \mathrm{d}v \, \mathrm{d}X + 2e^{2Q} X^+ X^- \, \mathrm{d}v^2 \tag{192}$$

where $Q$ is a known function of the dilaton $X$ and of an integration constant, the mass $M$. The Lorentz invariant combination $X^+ X^-$ also can be expressed as such a function, using the conserved Casimir inherent to the Poisson tensor (190). The metric (192) allows expressing worldsheet features in terms of conditions on the target space coordinates $X^I$. In the table below we summarize the relativistic interpretation of various loci on the worldsheet in terms of the signs of $X^\pm$.

| signs | $X^+ > 0$ | $X^+ < 0$ | $\boldsymbol{X^+ = 0}$ |
|---|---|---|---|
| $X^- > 0$ | anti-trapped | anti-normal | marginally anti-trapped |
| $X^- < 0$ | normal | trapped | marginally trapped |
| $\boldsymbol{X^- = 0}$ | marginally anti-trapped | marginally trapped | **extremal** |

For Kruskal-type of spacetimes "normal" refers to the outside region, "anti-normal" to the second outside region, "trapped" to the black hole region, "anti-trapped" to the white hole region, "marginally trapped" and "marginally anti-trapped" to the bifurcate Killing horizon and "extremal" to the bifurcation point. See Fig. 2 for a reminder.

A nice way of expressing what is special about extremal surfaces from a PSM perspective is to consider the action of relativistic boosts on the target space coordinates,

$$\delta_\lambda X = 0 \qquad \delta_\lambda X^\pm = \mp X^\pm \lambda \,. \tag{193}$$

Comparing with the table above, marginally trapped or anti-trapped surfaces are fixed lines (though not fixed-point lines) under boosts, since e.g. every locus $X^+ = 0$ is mapped to another locus where $X^+ = 0$. This provides a target space notion of marginally (anti-)trapped surfaces as fixed lines under boosts.

Similarly, by inserting the definition of extremal loci from the table above, we see that extremal surfaces are fixed points with respect to boosts: all the target space coordinates are invariant under boosts on the extremal locus.

**PSM definition of relativistic extremal surfaces.** Relativistic extremal surfaces are loci in the PSM target space that are fixed points under relativistic boosts.

This is the kind of property we were after. We have a definition of extremal surfaces as loci in the target space where the Poisson tensor is invariant under the gauge symmetries associated with boosts. Since we do not need the boosts to be relativistic for this definition to apply, it readily generalizes to Carroll boosts and thus allows defining Carroll extremal surfaces, which we shall do in the next Subsection. Before doing so, we translate back the definition above into a more familiar language.

In the second-order formulation, the target space coordinates $X^\pm$ do not exist but are replaced on-shell by directional derivatives of the dilaton field projected along the vielbein components.

$$X^\pm \approx \pm e_\pm^\mu \partial_\mu X \tag{194}$$

An extremal surface in the second-order formulation is thus a locus where the dilaton has a saddle point or an extremum.

$$\text{relativistic extremal surface:} \qquad e_\pm^\mu \partial_\mu X = 0 \qquad\qquad X > 0 \tag{195}$$

This is compatible with higher-dimensional intuition if 2d dilaton gravity is viewed as a dimensional reduction from higher-dimensional gravity. The condition of positive $X$ was added to eliminate ludicrous cases of fake extremal surfaces, which from a higher-dimensional perspective are spherical coordinate singularities, and from a 2d perspective, are singular loci where the effective gravitational coupling diverges.[10]

Similarly, we define (non-extremal) eternal black holes in terms of thermal and target space properties:

**PSM definition of relativistic eternal black holes.**   Relativistic eternal black holes are thermal states with finite entropy that have a relativistic extremal surface.

Thermality is needed to exclude extremal black holes, finite entropy is needed to exclude constant dilaton solutions, and the presence of a relativistic extremal surface is needed to ensure there is a special locus in target space that lies on the bifurcate Killing horizon of the worldsheet geometry. We are ready for the Carrollian generalization of extremal surfaces and black holes.

## 4.2   Carroll extremal surfaces in PSM formulation

Trying to mimic the relativistic classification of loci is less rich in the Carroll case, since the target space coordinate $X_\mathrm{P}$ does not appear on the right-hand side of the Carroll boost transformation laws (4), which we re-display here for convenience.

$$\delta_\lambda X = \delta_\lambda X_\mathrm{H} = 0 \qquad\qquad \delta_\lambda X_\mathrm{P} = X_\mathrm{H}\,\lambda \tag{196}$$

As expected, there is no natural notion of a Carroll horizon (since "everything moves with the speed of light"). However, there still is the notion of an extremal surface, as evident from (196), namely when $X_\mathrm{H} = 0$. The three different cases are summarized in the table below. (We label negative $X_\mathrm{H}$ as "normal" since, in most applications, $X_\mathrm{H}$ is negative between the asymptotic region and the extremal locus.)

| signs | $X_\mathrm{H} > 0$ | $X_\mathrm{H} < 0$ | $\mathbf{X_\mathrm{H} = 0}$ |
|---|---|---|---|
| | anti-normal | normal | **extremal** |

---

[10]A simple example of such a fake extremal surface is the centre of 4d Minkowski space in spherical coordinates, with $X = r^2$. The quantity $e_\pm^\mu \partial_\mu X \propto \pm \partial_r X = 2r$ vanishes at the origin $r = 0$.

Thus, we have a similar definition of Carroll extremal surfaces as in the relativistic case.[11]

**PSM definition of Carroll extremal surfaces.** Carroll extremal surfaces are loci in the PSM target space that are fixed points under Carroll boosts.

Note that every line of constant $X$ and $X_{\mathrm{H}}$ (but varying $X_{\mathrm{P}}$) is a fixed-line under Carroll boosts, so in that sense, every such line is "marginally (anti-)trapped" and the whole Carroll geometry could be viewed as a "horizon". On this "horizon" there can still be an exceptional point, the Carroll extremal surface, reminiscent of the bifurcation surface of relativistic black holes.

There is no Carrollian analogue of Carter–Penrose diagrams, but we can still draw diagrams similar to Fig. 2 to highlight different regions in the Carroll manifold with different signs of $X_{\mathrm{H}}$. Naturally, the diagrams in Fig. 3 are less rich in structure since there are fewer possibilities in the table above compared to the Lorentzian case. While we chose to draw the lines at 45° (as a reminder that null hypersurfaces have Carrollian structures), at this stage there is no significance to this angle. From a limiting perspective, one can understand these diagrams as emerging from infinite boosts of $t = \mathrm{const.}$ hypersurfaces ("wormholes") of Lorentzian black hole Carter–Penrose diagrams.

The three diagrams in Fig. 3 represent the same entity and emphasise different ways of boosting the constant time slice of the parent Carter–Penrose diagram. For instance, in the right diagram, the $t = \mathrm{const.}$ hypersurface in the parent Carter–Penrose diagram is boosted all the way to the future event horizon to both sides of the extremal surface. We shall elaborate on the relation to the wormhole picture in a higher-dimensional context in Section 6.

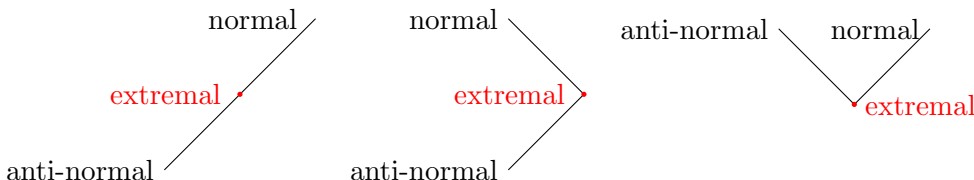

Figure 3: Three diagrams to visualize Carroll black holes

## 4.3 Carroll extremal surfaces in first- and second-order formulations

In the second-order version, the quantity $X_{\mathrm{H}}$ does not exist, but through the equations of motion (8) it is related on-shell to the directional derivative of the dilaton field projected onto the spatial inverse vielbein,

$$X_{\mathrm{H}} \approx -e^{\mu} \, \partial_{\mu} X \,. \tag{197}$$

Thus, in the second-order formulation, the criterion for an extremal surface is that the directional derivative of the dilaton field projected onto the spatial inverse vielbein vanishes.

$$\boxed{\text{Carroll extremal surface:} \qquad e^{\mu} \, \partial_{\mu} X = 0 \qquad\qquad X > 0} \tag{198}$$

---

[11]Perhaps also $|X_{\mathrm{H}}| = \infty$ has a similar interpretation, but since such loci typically are not part of the physical Carroll spacetime, we disregard this possibility here. These loci could appear at asymptotic boundaries, for instance, separating future and past null infinity.

This definition is analogous to the relativistic one (195) and is on-shell Carroll boost invariant, since $\delta_\lambda e^\mu \, \partial_\mu X = -\lambda v^\mu \, \partial_\mu X \approx X_{\mathrm{H}} \lambda \, v^\mu e_\mu = 0$. Note that one could add to (198) for free the condition $v^\mu \, \partial_\mu X = 0$ since $X$ does not depend on $t$ on-shell.

## 4.4 Carroll black holes

Equipped with our definition of extremal surfaces, we define Carroll black holes.

> **Definition of Carroll black holes.** Carroll black holes are C-thermal states with finite entropy that have a Carroll extremal surface.

In particular, we need the condition of finite entropy to exclude Carrollian constant dilaton solutions, which definitely should not be referred to as Carroll black holes.

In the next three Sections, we apply the results and definitions above to several examples.

# 5 Examples for 2d Carroll black holes

In this Section, we apply our general analysis to specific models, the Carroll JT model, the Carroll–Schwarzschild model, the Carroll CGHS model, and the Carroll Witten black hole. In each case, we ignore the constant dilaton vacua and focus exclusively on the linear dilaton sector.

The Carrollian thermodynamic quantities of solutions in the black hole sector of these models are listed in Table 1. The table also includes some other, more generic cases like the Carroll–Schwarzschild–Tangherlini solution and the Carroll $ab$-family.

| Model | $U(X)$ | $V(X)$ | $w(X)$ | $E$ | $T$ | $S$ | $C$ |
|-------|--------|--------|--------|-----|-----|-----|-----|
| CJT | $0$ | $\frac{X}{\ell^2}$ | $\frac{X^2}{2\ell^2}$ | $\frac{k}{2\pi}M$ | $\frac{\sqrt{2M}}{2\pi\ell}$ | $k\ell\sqrt{2M}$ | $2\pi k\ell^2 T$ |
| CS | $-\frac{1}{2X}$ | $\frac{\lambda^2}{4}$ | $\frac{\lambda^2}{2}\sqrt{X}$ | $\frac{k}{2\pi}M$ | $\propto \frac{1}{M}$ | $\propto M^2$ | $\propto -T^{-2}$ |
| CST | (36) | (36) | $\frac{\lambda^2}{2}X^{\frac{D-3}{D-2}}$ | $\frac{k}{2\pi}M$ | $\propto M^{\frac{1}{3-D}}$ | $\propto M^{\frac{D-2}{D-3}}$ | $\propto -T^{2-D}$ |
| CCGHS | $0$ | $\Lambda > 0$ | $\Lambda X$ | $\frac{k}{2\pi}M$ | $\frac{\Lambda}{2\pi}$ | $\frac{kM}{\Lambda}$ | $\infty$ |
| CWBH | $-\frac{1}{X}$ | $\frac{\lambda^2}{2}X$ | $\frac{\lambda^2}{2}X$ | $\frac{k}{2\pi}M$ | $\frac{\lambda^2}{4\pi}$ | $\frac{2kM}{\lambda^2}$ | $\infty$ |
| Cab | $-\frac{a}{X}$ | $\frac{B}{2}X^{a+b}$ | $\frac{B}{2(b+1)}X^{b+1}$ | $\frac{k}{2\pi}M$ | $\propto M^{\frac{b}{b+1}}$ | $\propto M^{\frac{1}{b+1}}$ | $\frac{k}{b}\left(\frac{4\pi T}{B}\right)^{\frac{1}{b}}$ |

Table 1: Carrollian thermodynamic quantities for the Carroll JT model (CJT), Carroll–Schwarzschild (CS), Carroll–Schwarzschild–Tangherlini (CST) in $D$ spacetime dimensions, Carroll CGHS (CCGHS), Carroll Witten black hole (CWBH), and the Carroll $ab$-family (Cab). As $w'' = 0$ for CCGHS and CWBH the specific heat diverges for these models. In some expressions, we left out state-independent prefactors for brevity which is denoted by $\propto$.

In the first two Subsections 5.1-5.2, we show the spectrum, thermodynamics, and an example for boundary conditions of the CJT model. In Subsection 5.3, we present the 2d perspective of the Carroll–Schwarzschild black hole. In Subsection 5.4, we investigate the CCGHS model. In Subsection 5.5, we present the Carroll–Witten black hole.

## 5.1 Carroll JT model

The Jackiw–Teitelboim (JT) model [49,50] was the first 2d model of gravity. It is particularly elegant, as it allows a reformulation as non-abelian BF theory [62,63], in contrast to nearly all other 2d dilaton gravity models. All its solutions are locally (A)dS$_2$ so that JT gravity is tailor-made for a holographic description [77–83]. Especially the SYK/JT correspondence [84–87] has reinvigorated the interest in JT gravity and its holographic description. Due to its simple BF formulation, the JT model was the starting point for Carrollian limits of 2d dilaton gravity [43]. Here, we summarize the key results for the Carroll JT (CJT) model, and in Subsection 5.2, we discuss boundary conditions for CJT.

The CJT model is given by the Lagrangian (3) with the potentials

$$U_{\mathrm{CJT}}(X) = 0 \qquad V_{\mathrm{CJT}}(X) = \frac{1}{\ell^2}\, X\,. \tag{199}$$

The function $w$ defined in (130) for CJT is given by

$$w_{\mathrm{CJT}}(X) = \frac{X^2}{2\ell^2}\,. \tag{200}$$

Applying our general analysis of Section 3 to the choice (199) yields the linear dilaton solutions (we fix the integration constant coming from integrating (133) without loss of generality by a shift of the origin of the spatial coordinate $r$, and we take the positive branch of the square-root function)

$$X = \frac{1}{2}\, e^{r/\ell} + M\ell^2\, e^{-r/\ell} \qquad\qquad \omega = \frac{X}{\ell^2}\, \mathrm{d}t \tag{201a}$$

$$X_{\mathrm{H}} = -\frac{1}{2\ell}\, e^{r/\ell} + M\ell\, e^{-r/\ell} \qquad\qquad \tau = -X_{\mathrm{H}}\, \mathrm{d}t \tag{201b}$$

$$X_{\mathrm{P}} = 0 \qquad\qquad e = \mathrm{d}r\,. \tag{201c}$$

Translating the 1-forms into second-order notation, the solution above reads

$$\mathrm{d}s^2 = \mathrm{d}r^2 \qquad v = \frac{2\ell\, e^{-r/\ell}}{2M\ell^2\, e^{-2r/\ell} - 1}\, \partial_t \qquad X = \frac{1}{2}\, e^{r/\ell} + M\ell^2\, e^{-r/\ell}\,. \tag{202}$$

In Fig. 4 we depict a constant $X_{\mathrm{P}}$ slice of the PSM target space associated with the CJT model.[12] The spectrum of CJT falls into three classes, depending on the sign of the mass parameter $M$:

- $M < 0$: no Carroll black hole, since $X_{\mathrm{H}} < 0$ everywhere, reminiscent of the global AdS$_2$ solution of JT

- $M = 0$: limiting case, where $X_{\mathrm{H}} \to 0$ as $r \to -\infty$, reminiscent of the Poincaré horizon of the massless JT solution

- $M > 0$: Carroll black holes, since $X_{\mathrm{H}} = 0$ has the solution $X = \ell\sqrt{2M}$ or, equivalently, $r = \frac{\ell}{2}\ln(2M\ell^2)$, reminiscent of black hole solutions of JT

We focus on the positive mass sector since it features Carroll extremal surfaces. Furthermore, to have positive entropy we restrict to the branches with $X > 0$.

Energy, entropy, temperature, and specific heat of these solutions are given in Table 1, and the first law is satisfied, as shown in Section 3.3. Expressing the entropy as a

---

[12]For positive mass, at the Carroll extremal surface $X = \ell\sqrt{2M}$ the solution can be joined to one where $X_{\mathrm{H}} \to -X_{\mathrm{H}}$ and hence $v \to -v$.

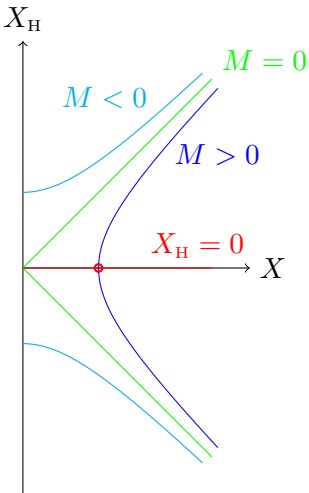

Figure 4: Target space picture of Carroll JT by plotting the level sets of (143), restricted to the region $X \geq 0$. Extremal points are red circles and exist only for $M > 0$. The solutions given in (201) cover the lower half of this diagram.

function of the energy shows a relation similar to the Cardy formula for a chiral half of a 2d conformal field theory,

$$S = \frac{\pi^2 c\, T}{3} = 2\pi \sqrt{\frac{c\, E}{6}} \tag{203}$$

provided the central charge is chosen as

$$c = \frac{6k\ell^2}{\pi} \,. \tag{204}$$

This is again reminiscent of the relativistic case [70].

## 5.2  Example of boundary conditions for CJT

One can interpret (202) as radial Gaussian coordinates and provide a "Fefferman–Graham" expansion for the vector field and the dilaton

$$v = 2\ell e^{-r/\ell} \big( -1 + \mathcal{O}(e^{-2r/\ell}) \big)\, \partial_t \qquad\qquad X = \frac{1}{2}\, e^{r/\ell} \big( 1 + \mathcal{O}(e^{-2r/\ell}) \big) \tag{205}$$

where the leading terms are fixed, and the subleading terms contain state-dependent information. Similarly to JT gravity, there are numerous inequivalent choices for boundary conditions [70]. It is not our intention to exhaustively discuss the possibilities for CJT gravity. Instead, we provide just one example for boundary conditions and leave a more comprehensive study for future work.

The Brown–Henneaux-like boundary conditions

$$X = \frac{1}{2}\, e^{r/\ell} + M(t)\, \ell^2 e^{-r/\ell} \qquad\qquad \omega = \frac{1}{2\ell^2}\, e^{r/\ell}\, \mathrm{d}t + \mathcal{O}(e^{-r/\ell}) \tag{206a}$$

$$X_{\mathrm{H}} = -\frac{1}{2\ell}\, e^{r/\ell} + M(t)\, \ell\, e^{-r/\ell} \qquad\qquad \tau = \frac{1}{2\ell}\, e^{r/\ell}\, \mathrm{d}t + \mathcal{O}(e^{-r/\ell}) \tag{206b}$$

$$X_{\mathrm{P}} = 0 \qquad\qquad\qquad\qquad\qquad e = \mathrm{d}r \tag{206c}$$

with $\delta M \neq 0$ are preserved by the gauge transformations (4)-(6) with gauge parameters $\lambda_{\mathrm{P}} = 0$ and

$$\lambda = e^{r/\ell}\frac{\eta}{\ell} + 2\ell M(t)\eta\, e^{-r/\ell} \qquad\qquad \lambda_{\mathrm{H}} = e^{r/\ell}\eta - 2\ell^2 M(t)\eta\, e^{-r/\ell} \qquad (207)$$

where $\eta$ is the transformation parameter. The equations of motion (8) are solved by the field configuration (206), up to subleading terms (which can be determined in closed form, if desired). On-shell the mass function $M(t)$ is given by the Casimir $M = \frac{X^2}{2\ell^2} - \frac{X_{\mathrm{H}}^2}{2}$.

The variation of the boundary charges

$$\delta\mathcal{Q}[\lambda_I] = \frac{k}{2\pi}\left(\lambda\,\delta X + \lambda_{\mathrm{H}}\,\delta X_{\mathrm{H}} + \lambda_{\mathrm{P}}\,\delta X_{\mathrm{P}}\right) \qquad (208)$$

in the present case can be integrated (in field space) to a single boundary charge

$$\mathcal{Q}[\eta] = \frac{k\ell}{\pi}\,\eta\,M(t) \qquad (209)$$

which is finite as the radial coordinate approaches the asymptotic boundary, $r \to \infty$. On-shell it is also conserved, $\partial_t \mathcal{Q}[\eta] \approx 0$. We assumed here a slicing of the phase space where $\eta$ is state-independent (see, e.g., [64,88] for a discussion of different phase space slicings in Lorentzian 2d dilaton gravity).

The asymptotic symmetry algebra trivially is abelian in the present case since we have only one boundary charge, namely the Casimir $M$.

$$\{\mathcal{Q}[\eta_1],\,\mathcal{Q}[\eta_2]\} \approx \delta_{\eta_2}\mathcal{Q}[\eta_1] \approx \frac{k\ell}{\pi}\,\eta_1\,\delta_{\eta_2}M = 0 \qquad (210)$$

### 5.3 Carroll–Schwarzschild black hole, 2d perspective

As reviewed in Section 2.1.2, spherical reduction of Einstein gravity leads to a specific 2d dilaton gravity model, the solutions of which reproduce the Schwarzschild black hole. There is an expansive history of spherically reduced gravity [89–92] that predates the developments of 2d dilaton gravity. Here, we consider the Carrollian limit of the Schwarzschild black hole from a 2d perspective. See Section 6 for a 4d perspective.

The spherically reduced Carroll–Schwarzschild (CS) model is given by the potentials (36), which for $D = 4$ are

$$U_{\mathrm{CS}}(X) = -\frac{1}{2X} \qquad\qquad V_{\mathrm{CS}}(X) = \frac{\lambda^2}{4}\ . \qquad (211)$$

The functions $w_{\mathrm{CS}}$ and $e^{Q_{\mathrm{CS}}}$ are

$$e^{Q_{\mathrm{CS}}} = \frac{1}{2\sqrt{X}} \qquad\qquad w_{\mathrm{CS}}(X) = \frac{\lambda^2}{4}\sqrt{X} \qquad (212)$$

where we chose the integration constant of the second integral in (142) accordingly. This model is described by a target space diagram given in Fig. 5. The solutions never take negative values of the dilaton. Moreover, the black hole sector of the model is given by $M > 0$ as all other solutions do not lead to states with finite entropy $S \sim X_{\mathrm{ext.}}$. Let us choose $\lambda = 2$ for convenience. This implies that the dilaton measures the surface radius as seen from the higher-dimensional setting, i.e., the spherical part of the 4d metric reads

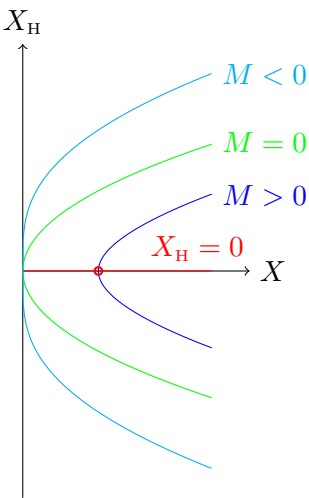

Figure 5: Target space picture of spherically reduced Carroll–Schwarzschild black hole. Extremal points are red circles and exist only for $M > 0$. The other symplectic leaves do not exhibit such points as for $M = 0$ the point would be at $X = 0$ and for $M < 0$ the leaves do not contain points with $X_\mathrm{H} = 0$ at all (they are not simply connected). The black hole sector is thus given by $M > 0$. The solutions (213)-(215) describe the lower half of the diagram.

$X\, \mathrm{d}\Omega^2_{S^2}$ (see also (31), (34)). Applying our analysis from Section 3 yields

$$X_\mathrm{H} = -\sqrt{4X - 4M\sqrt{X}} \qquad\qquad \omega = \frac{2\sqrt{X} - M}{2X}\,\mathrm{d}t \qquad (213)$$

$$X_\mathrm{P} = 0 \qquad\qquad \tau = -\frac{X_\mathrm{H}}{2\sqrt{X}}\,\mathrm{d}t \qquad (214)$$

$$e = \mathrm{d}r \qquad (215)$$

and a 2d Carrollian curvature scalar

$$R = 4e^\mu v^\nu \partial_{[\mu}\hat\omega_{\nu]} = \frac{2M}{X^{\frac{3}{2}}} \qquad\qquad \hat\omega = \omega - U(X)X_\mathrm{H}\tau \qquad (216)$$

where $\hat\omega$ is defined as the torsion-free part of the Carrollian connection [see (19)]. The Carrollian second-order variables read

$$v = -\frac{1}{\sqrt{1 - \frac{M}{\sqrt{X}}}}\partial_t \qquad\qquad h = \frac{\mathrm{d}X^2}{4X - 4M\sqrt{X}} \qquad (217)$$

where the vector field is normalized asymptotically, $\lim_{X\to\infty} v = -\partial_t$. For simplicity, in these solutions, the ambiguity in the torsion-free spin connection was fixed to $\rho = 0$. To bring this into a more familiar form, we can define the radial coordinate

$$\mathsf{r}^2 = X \qquad (218)$$

which together with the Schwarzschild mass $m = \frac{M}{2}$ leads to

$$v = -\frac{1}{\sqrt{1 - \frac{2m}{\mathsf{r}}}}\partial_t \qquad\qquad h = \frac{\mathrm{d}\mathsf{r}^2}{1 - \frac{2m}{\mathsf{r}}}\,. \qquad (219)$$

The Carrollian thermodynamic quantities for the Carroll–Schwarzschild black hole

$$E = \frac{k}{\pi}\, m \qquad\qquad T = \frac{1}{8\pi m} \qquad\qquad S = 4km^2 \qquad (220)$$

satisfy the first law,[13]

$$\delta E = T\,\delta S\ . \qquad (221)$$

Generalizing Carroll–Schwarzschild to Carroll–Schwarzschild–Tangherlini is straightforward, and the main results are summarized in Table 1.

In Section 6, we provide the 4d perspective on these solutions.

## 5.4   Carroll CGHS

The Callan–Giddings–Harvey–Strominger (CGHS) model [93] is a 2d toy model for black hole evaporation. It consists of a Lorentzian 2d dilaton gravity action with potentials $U = 0$, $V = \Lambda = $ const. plus some minimally coupled scalar fields as carriers of the Hawking quanta. In our work, we always neglect interactions with matter, so when we refer to the CGHS model or its Carrollian counterpart, we solely mean the geometric part of the model without matter. Besides the JT model, the CGHS model is arguably the simplest 2d dilaton gravity model. A more precise version of this statement is that only the JT and the CGHS model permit a reinterpretation of the corresponding PSM as non-abelian BF theory. This is the main reason why the CGHS model was the first one to receive a holographic interpretation [94] after the JT model.

The Carrollian limit of the CGHS model (CCGHS) has the same potentials

$$U_{\text{CCGHS}} = 0 \qquad\qquad V_{\text{CCGHS}} = \Lambda = \text{const.} > 0\,. \qquad (222)$$

The solutions of the linear dilaton sector are

$$X = \frac{\Lambda}{2}r^2 + \frac{M}{\Lambda} \qquad\qquad\qquad \omega = \Lambda\,\mathrm{d}t \qquad (223)$$

$$X_{\text{H}} = -\Lambda r \qquad\qquad\qquad\qquad \tau = -X_{\text{H}}\,\mathrm{d}t \qquad (224)$$

$$X_{\text{P}} = 0 \qquad\qquad\qquad\qquad\qquad e = \mathrm{d}r \qquad (225)$$

leading to a flat Carrollian spacetime, i.e., $R = 0$. Here, the radial coordinate was fixed such that $r = 0$ corresponds to $X_{\text{H}} = 0$. Investigating the spectrum of this model shows that Carroll extremal surfaces exist only for positive values of $M$ (see Fig. 6). The various Carrollian thermodynamic quantities of these solutions are given in Table 1. As the temperature is fixed to a single specific value in terms of the model-dependent constant $\Lambda$, the specific heat diverges.

## 5.5   Carroll Witten black hole

The Witten black hole [51–53] emerges from 2d string theory. From the worldsheet perspective, it is described by an $\mathrm{SL}(2,\mathbb{R})/U(1)$ gauged WZW-model. Interpreting the vanishing of the $\beta$-functions of this conformal field theory as target space equations of motion yields as target space action a Lorentzian 2d dilaton gravity model with potentials $U = -\frac{1}{X}$ and $V = \frac{\lambda^2}{2}X$, where $\lambda^2 \propto 1/\alpha'$ with the inverse string tension $\alpha'$. As common in the literature, we use the phrase "Witten black hole" as a label for the conformal field theory, the target space theory, and the positive mass spectrum of solutions to the latter. The Euclidean continuation of the Witten black hole is the famous cigar geometry,

---

[13]From a 4d perspective, the coupling constant $k$ is given by $\pi/G_M$ in units where $\lambda = 2$, see the Carrollian version of Eq. (37).

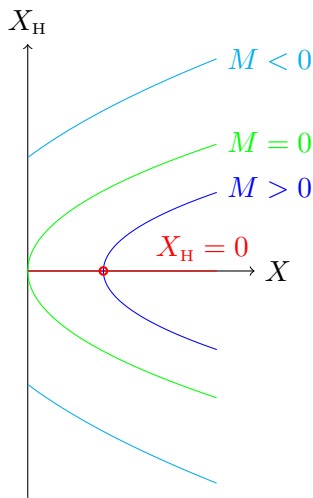

Figure 6: Target space picture of Carroll CGHS with $X_{\rm P} = 0$, restricted to the region $X \geq 0$. Extremal points are red circles and exist only for $M > 0$. The lower half is described by the solutions (223)-(225).

$\mathrm{d}s^2 = \mathrm{d}r^2 + \tanh^2 r \, \mathrm{d}\tau^2$. For more details on the Witten black hole, see Section 2.1.2 in [55] and Refs. therein.

The Carroll limit of the Witten black hole features the same potentials

$$U_{\rm CWBH} = -\frac{1}{X} \qquad\qquad V_{\rm CWBH} = \frac{\lambda^2}{2} X > 0 \tag{226}$$

and is referred to as Carroll Witten black hole. Analogously to its Lorentzian avatar (see e.g. [95]), it emerges as $D \to \infty$ limit of the CST black hole and is conformally related to the CCGHS model by a dilaton-dependent Weyl rescaling.

In the linear dilaton sector, the CWBH solutions

$$X = \frac{2M}{\lambda^2} \cosh^2 \frac{\lambda r}{2} \qquad\qquad \omega = \left(\lambda^2 - \frac{M}{X}\right) \mathrm{d}t \tag{227}$$

$$X_{\rm H} = -\sqrt{\lambda^2 X^2 - 2MX} \qquad\qquad \tau = -\frac{X_{\rm H}}{X} \, \mathrm{d}t \tag{228}$$

$$X_{\rm P} = 0 \qquad\qquad e = \mathrm{d}r \tag{229}$$

lead to the same thermodynamical behaviour as the CCGHS model. In fact, all thermodynamic formulas are equivalent for CCGHS and CWBH upon replacing $\Lambda \to \frac{\lambda^2}{2}$.

# 6 Carroll–Schwarzschild black hole, 4d perspective

In this Section, we elaborate on the 4d perspective of the Carroll—Schwarzschild black hole and the associated wormhole picture. While the following discussion easily generalizes to higher dimensions (Section 5), we focus, for clarity, on $3 + 1$ dimensions.

The Schwarzschild line element is given by

$$\mathrm{d}s^2 = -\left(1 - \frac{\mathsf{r}_s}{\mathsf{r}}\right) c^2 \, \mathrm{d}t^2 + \frac{\mathrm{d}\mathsf{r}^2}{1 - \frac{\mathsf{r}_s}{\mathsf{r}}} + \mathsf{r}^2 \, \mathrm{d}\Omega_{S^2}^2 \tag{230}$$

where $\mathsf{r}_s = \frac{2mG}{c^2}$ and $\mathrm{d}\Omega_{S^2}^2$ is the metric on the round 2-sphere.

In order to take the magnetic Carroll limit, we again rescale Newton's constant as $G_M = Gc^{-4}$ and we keep $G_M$ and $r_s$ fixed as we expand around $c = 0$. The general $c = 0$ expansion of any Lorentzian metric takes the form [65]

$$\mathrm{d}s^2 = h_{MN}\, \mathrm{d}x^M\, \mathrm{d}x^N + c^2 \left(-\tau_M \tau_N + \Phi_{MN}\right) \mathrm{d}x^M\, \mathrm{d}x^N + \mathcal{O}(c^4) \tag{231}$$

where the Carroll metric $h_{MN}$ has signature $(0, +, +, +)$ and $v^M v^N \Phi_{MN} = 0$. We define the Carroll vector field $v^M$ to obey the usual conditions $v^M \tau_M = -1$ and $v^M h_{MN} = 0$. (See Appendix A for details on global and local Carroll symmetries.) One can also find $v^M$ from the leading-order term in the $c = 0$ expansion of the inverse metric.

The magnetic limit of the Schwarzschild black hole [65, 96]

$$v = -\frac{1}{\sqrt{1 - \frac{r_s}{r}}}\, \partial_t \qquad\qquad h = \frac{\mathrm{d}r^2}{1 - \frac{r_s}{r}} + r^2\, \mathrm{d}\Omega_{S^2}^2 \tag{232}$$

is the lifted version of (219). This configuration is a solution of magnetic Carroll gravity [96, 97]. The extension with a non-vanishing cosmological constant was described in [98].

It is instructive to rewrite this configuration in terms of isotropic coordinates obtained by the (double cover) coordinate transformation $r \mapsto \rho = \rho(r)$ given by

$$r = \frac{r_s}{4}\left(\rho + \frac{1}{\rho} + 2\right) \tag{233}$$

resulting in the Carrollian wormhole geometry

$$v = -\frac{\rho + 1}{\rho - 1}\, \partial_t \qquad\qquad h = r_s^2 \left(\frac{(\rho + 1)^2}{4\rho^2}\right)^2 \left(\mathrm{d}\rho^2 + \rho^2\, \mathrm{d}\Omega_{S^2}^2\right). \tag{234}$$

The spatial Carroll metric $h$ is $\rho \to 1/\rho$ symmetric, and $v$ changes sign under this map, corresponding to the well-known fact that the Killing time runs opposite in the universe on the other side of the wormhole (see Fig. 7).

Let us scan for Carroll extremal surfaces, cf. Section 4. By definition, they satisfy $e_a^M \partial_M X = 0$, which is the condition that these surfaces are invariant under any linear deviations, regardless of the direction. The dilaton $X$ is the surface area of the 2-spheres that foliate our spherically symmetric spacetime. For (232) it is given by $X = r^2$ [with $\lambda = 2$ in (34)]. This means that, using the conventions of Section 2.2.3, only the radial part of the inverse vielbein $e_1^M \partial_M =: e^M \partial_M$ leads to a nontrivial condition

$$e^M \partial_M X = \left(1 - \frac{r_s}{r}\right) \partial_r X \overset{!}{=} 0. \tag{235}$$

The Carroll extremal surface is at $r = r_s$ or, equivalently, at $\rho = 1$ (see Fig. 7).

From the 4d perspective, it is natural to assign the dilaton the length dimension $[X] = 2$, which implies $[r] = 1$ and $[k] = -2$ (cf. the discussion in Section 3.4). The relativistic entropy, temperature, and energy of the Schwarzschild black hole are given by

$$S_{\mathrm{rel}} = \frac{\pi c^3 r_s^2}{\hbar G} \qquad\qquad T_{\mathrm{rel}} = \frac{\hbar c}{4\pi r_s} \qquad\qquad E_{\mathrm{rel}} = \frac{c^4}{2G} r_s \tag{236}$$

where we restored all conversion factors except for Boltzmann's constant, which we fix to one. Expanding in powers of $c$ while keeping $G_M = c^{-4}G$ and $r_s$ fixed leads to the leading-order Carrollian quantities

$$S = \frac{\pi r_s^2}{\hbar c G_M} \qquad\qquad T = \frac{\hbar c}{4\pi r_s} \qquad\qquad E = \frac{1}{2G_M} r_s. \tag{237}$$

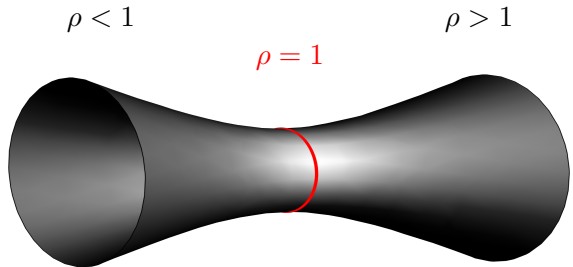

$\rho < 1$                    $\rho > 1$

$\rho = 1$

Figure 7: Sketch of spatial Carroll wormhole geometry (234). It corresponds to the spatial wormhole geometry of the maximally extended Schwarzschild black hole that cuts through the bifurcation sphere. In red, where $\rho = 1$ ($\mathsf{r} = \mathsf{r}_s$), we have encircled the Carroll extremal surface at the wormhole's throat.

The results (237) coincide with the general results in 2d derived in Section 3, using the 2d-4d dictionary (note that our choices imply $e^Q = \hbar c/(2\sqrt{X})$)

$$k = \frac{\pi}{\hbar c G_M} \qquad\qquad X = \mathsf{r}^2 \qquad\qquad X_{\min} = \mathsf{r}_s^2 \qquad\qquad w(X) = \hbar c \sqrt{X}\,. \tag{238}$$

These dimensions work as required since $G_M$ has dimensions of metre/Joule, $\hbar c$ metre times Joule, $k$ has dimension of one over metre squared, and $X$ is measured in square metres. So entropy is dimensionless, while temperature and energy are measured in Joule.

We show next that the expressions (237) can be computed using 4d geometric arguments that are similar to what one does in general relativity. As shown above, the Carroll entropy is proportional to the area of the wormhole's throat. At this locus we have $h|_{\rho=1} = \mathsf{r}_s^2\, \mathrm{d}\Omega_{S^2}^2$. As discussed above and in Section 3, we ensured that $S$ is dimensionless.

We next turn to temperature. The 4d Carroll boost and rotation connections $\omega^a$ and $\omega^{ab} = -\omega^{ba}$ are solutions to the Cartan zero torsion-equations

$$\mathrm{d}\tau + \omega^a \wedge e^a = \mathrm{d}e^a + \omega^{ab} \wedge e^b = 0\,. \tag{239}$$

For the Carroll wormhole solution (232), we choose the vielbeins

$$\tau = f(\mathsf{r})\, \mathrm{d}t \qquad\qquad e^1 = f^{-1}(\mathsf{r})\, \mathrm{d}\mathsf{r} \qquad\qquad e^l = \mathsf{r}\, \bar{e}^l \tag{240}$$

where $l = 2, 3$ correspond to the 2-sphere tangent space directions, $\bar{e}^l$ are round unit 2-sphere vielbeins, and we defined $f(\mathsf{r}) = (1 - \mathsf{r}_s/\mathsf{r})^{1/2}$. The most general solution to these equations is

$$\omega^1 = f'\tau + \rho\, e^1 + \rho^l e^l \qquad\qquad \omega^l = \rho^l e^1 + \rho^{lm} e^m \qquad\qquad \omega^{1l} = f\bar{e}^l \tag{241}$$

and $\omega^{lm} = \bar{\omega}^{lm}$ is the connection on the unit 2-sphere.

We next consider the pullback of (239) onto the 2d submanifold obtained by fixing a point on the 2-sphere. In order to avoid clutter, we denote the pullbacks of the vielbeins by the same symbols. If the 2-sphere has coordinates $\theta, \phi$ we are considering the manifold $\theta = \theta_0$ and $\phi = \phi_0$ where $\theta_0$ and $\phi_0$ are constants. The equations (239) on this submanifold become

$$\mathrm{d}\tau + \omega^1 \wedge e^1 = \mathrm{d}e^1 = 0 \tag{242}$$

with $\tau = f(\mathsf{r})\, \mathrm{d}t$, $e^1 = f^{-1}\, \mathrm{d}\mathsf{r}$ and $\omega^1 = f'\tau + \rho\, e^1$.

To define the Carroll temperature $T$, we first Wick rotate by the prescription

$$t = it_{\mathrm{w}} \qquad\qquad \tau = i\tau_{\mathrm{w}} \qquad\qquad \omega^1 = i\omega_{\mathrm{w}}^1 \qquad\qquad v = iv_{\mathrm{w}} \tag{243}$$

where it is understood that the conversion factor has already been absorbed such that the length dimensions are $[\tau_{\mathrm{W}}] = [t_{\mathrm{W}}] = 1$ and $[\omega_{\mathrm{W}}] = 0$ (again, see Section 3.4). The Wick rotation does not change the signature of the geometry and the holonomy of the manifold (unlike in general relativity) but it allows us to consider (242) for a periodic $t_{\mathrm{W}}$. Now, $t_{\mathrm{W}}$ has the dimension of a length such that it can be compactified with period $\hbar c/T$. In the Wick rotated setting, (242) becomes $\mathrm{d}\tau_{\mathrm{W}} + \omega_{\mathrm{W}}^1 \wedge e^1 = 0$, where $\tau_{\mathrm{W}} = f(\mathsf{r})\,\mathrm{d}t_{\mathrm{W}}$ has dimensions of length and $\omega_{\mathrm{W}}^1 = f'\tau_{\mathrm{W}} + \bar{\rho}e^1$ is dimensionless. The 2d geometry described by $t_{\mathrm{W}} \sim t_{\mathrm{W}} + \hbar c/T$ and $\mathsf{r}_s < \mathsf{r} < \mathsf{r}_c$ where $\mathsf{r}_c$ is some arbitrary number is a cigar-like geometry that we will denote by $\Sigma$. The boundary of $\Sigma$ is given by the circle at $\mathsf{r} = \mathsf{r}_c$. The Carroll boost connection $\omega_{\mathrm{W}}^1$ contains an undetermined function $\bar{\rho}$. We assume that $\bar{\rho}$ is globally well-defined on the cigar geometry so that it is periodic in $t_{\mathrm{W}}$. Then, a direct calculation tells us

$$\int_\Sigma \mathrm{d}\omega_{\mathrm{W}}^1 - \int_{\partial\Sigma} \omega_{\mathrm{W}}^1 = \frac{1}{2\mathsf{r}_s}\frac{\hbar c}{T} \tag{244}$$

where $\partial\Sigma$ is the circle at $\mathsf{r} = \mathsf{r}_c$. The bulk orientation is chosen such that $\mathrm{d}t_{\mathrm{W}} \wedge \mathrm{d}r =: \mathrm{d}t_{\mathrm{W}}\,\mathrm{d}r$ and the boundary orientation is induced similarly to Section 2.3.5. The left-hand side is $2\pi$ times the Euler character $\chi$ of $\Sigma$ which is topologically a disk so that $\chi = 1$. This procedure recovers the result for temperature announced in (237).

In [96] the energy $E$ of the Carroll solution discussed here was computed. The result is the same as for the Schwarzschild black hole, namely (in magnetic Carroll units)

$$E = \frac{\mathsf{r}_s}{2G_M} = 2TS \tag{245}$$

in agreement with (237). Of course, the first law

$$\delta E = T\,\delta S \tag{246}$$

is obeyed.

All the thermodynamical relations above follow immediately from the general analysis of Section 3, using the 2d-4d dictionary (238).

# 7 Charged and rotating Carroll black holes

All examples so far can be understood entirely in terms of 2d Carroll dilaton gravity or one of its dimensional uplifts to higher dimensions. In this Section, we go beyond this case by considering charged or rotating Carroll black holes (mostly from a dimensionally reduced, 2d, perspective), where a generalization to 2d Carroll–Maxwell dilaton gravity is required.

## 7.1 General remarks on charged Carroll black holes in 2d

In the PSM formulation, adding a Maxwell field amounts to adding another target space coordinate $Y$ and adding to the Poisson tensor an extra row and column of zero entries. The potential in (3) can depend on this additional target space coordinate as well, and the Lagrange 2-form acquires an additional term $Y\,\mathrm{d}A$, where $A = A_\mu\,\mathrm{d}x^\mu$ is the Maxwell gauge field 1-form.

$$\mathcal{L} = Y\,\mathrm{d}A + X\,\mathrm{d}\omega + X_{\mathrm{H}}\left(\mathrm{d}\tau + \omega \wedge e\right) + X_{\mathrm{P}}\,\mathrm{d}e + \mathcal{V}(X, X_{\mathrm{H}}, Y)\,\tau \wedge e \tag{247}$$

The additional $U(1)$ gauge symmetry generated by some transformation parameter $\Lambda$ acts trivially on all fields except on the Maxwell gauge field, $\delta_\Lambda A = \mathrm{d}\Lambda$. The equations of

motion (8) are essentially unchanged, with the replacement $\mathcal{V}(X, X_\mathrm{H}) \to \mathcal{V}(X, X_\mathrm{H}, Y)$. There are two additional equations of motion from varying with respect to $Y$ and $A$:

$$\delta Y: \qquad\qquad \mathrm{d}A = -\frac{\partial \mathcal{V}(X, X_\mathrm{H}, Y)}{\partial Y}\, \tau \wedge e \qquad\qquad (248)$$

$$\delta A: \qquad\qquad \mathrm{d}Y = 0 \qquad\qquad (249)$$

The quantity $Y$ on-shell is a second conserved Casimir,

$$Y = q = \text{const.} \qquad\qquad (250)$$

and physically corresponds to a conserved $U(1)$ charge (though in some applications, it might have a different interpretation, e.g., as angular momentum in higher dimensions).

If the Lagrange 2-form (247) emerges from some Carrollian limit, it could happen that the first and last terms are multiplied by some powers of the speed of light $c$. In that case, we can always first rescale $Y$ with an appropriate factor of $c$ to eliminate $c$ from the last term and then rescale $A$ with an appropriate factor of $c$ to eliminate $c$ from the first term. Thus, without loss of generality, we assume there are no explicit factors of $c$ in the Lagrange 2-form (247).

Solving the equations of motion can be done exactly as in Section 2. The solutions for the spatial metric and the temporal vector field will, in general, depend not only on the mass parameter $M$ but also on the $U(1)$ charge $q$. As a consequence of this dependence, there can be BPS-like bounds and extremality conditions. (Since we already use the word "extremal" to denote Carroll extremal surfaces, we call the confluent case "degenerate" instead.) Relatedly, new constant dilaton vacua can emerge and typically have an interpretation as "near horizon extremal geometries". (To clarify also here the vocabulary, we refer to such geometries as "near-Carroll-extremal-surface degenerate geometries" in a Carrollian context.) Other than these marginal changes, our general discussion of Section 2 applies.

A prototypical form of the potential is given by

$$\mathcal{V}(X, X_\mathrm{H}, Y) = \hat{\mathcal{V}}(X, X_\mathrm{H}) - \frac{Y^2}{4F(X)}\,. \qquad\qquad (251)$$

In this case, the charge $q$ on-shell is related to the electric field, $E = * \mathrm{d}A$, and the dilaton,

$$q = Y = 2F(X) * \mathrm{d}A = 2F(X)\, E \qquad\qquad (252)$$

where we used the Carroll–Hodge-$*$ relation $*(\tau \wedge e) := 1$. For a definition of this operator, we refer to [99]. Charge conservation implies

$$\mathrm{d}\big(F(X) * \mathrm{d}A\big) = 0\,. \qquad\qquad (253)$$

Integrating out the scalar field $Y$ by its own equation of motion yields a (non-minimally coupled) Maxwell term in the Lagrange 2-form (with the usual expression for the field strength, $F_{\mu\nu} = \partial_\mu A_\nu - \partial_\nu A_\mu$).

$$\mathcal{L} = X\, \mathrm{d}\omega + X_\mathrm{H}\big(\mathrm{d}\tau + \omega \wedge e\big) + X_\mathrm{P}\, \mathrm{d}e + \hat{\mathcal{V}}(X, X_\mathrm{H})\, \tau \wedge e + F(X)\, \underbrace{(* \mathrm{d}A)\, \mathrm{d}A}_{\sim F_{\mu\nu}F^{\mu\nu}\, \text{vol}} \qquad (254)$$

Consistently, varying (254) with respect to the Maxwell connection $A$ yields the equation of motion (253). For models that come from a dimensional reduction of higher-dimensional Einstein–Maxwell theories the coupling function $F(X)$ typically is linear in the dilaton since the Maxwell term gets the same volume factor as the curvature term $X\, \mathrm{d}\omega$.

Finally, we note that often it is sufficient to consider the charge $q$ as a parameter in the action rather than as a constant of motion. In that case, one can use the potential (251) with $Y$ replaced by its on-shell value $q$.

## 7.2  Carroll–Reissner–Nordström

There are different paths to obtaining CRN black holes. We found it simplest to first reduce spherically Schwarzschild to 2d, then take the Carroll limit, and finally, add a Maxwell field. This means we take the Schwarzschild results for the potential $\hat{\mathcal{V}}$ and set $F(X) = X$,

$$\mathcal{V}_{\mathrm{CRN}}(X, X_{\mathrm{H}}, Y) = \frac{\lambda^2}{4} + \frac{X_{\mathrm{H}}^2}{4X} - \frac{Y^2}{4X} \,. \tag{255}$$

Without loss of generality, we set $\lambda = 2$.

In the coordinates introduced in Section 2, the CRN solution is given by

$$\mathrm{d}s^2 = \mathrm{d}r^2 = \frac{\mathrm{d}X^2}{X_{\mathrm{H}}^2} \qquad\qquad v = \frac{2\sqrt{X}}{X_{\mathrm{H}}}\, \partial_t \tag{256}$$

and

$$X_{\mathrm{H}} = \pm\sqrt{4X + 4\,q_e^2 - 8m\sqrt{X}} \,, \tag{257}$$

where $q_e = \frac{q}{2}$ is the electric charge and $m = \frac{M}{2}$ is the mass. We again chose the integration constant in (142) such that $e^{-Q} = 2\sqrt{X}$ to achieve an asymptotic normalization of the vector field, $\lim_{X\to\infty} v = -\partial_t$. The solution for the gauge field follows from (248),

$$\mathrm{d}A = \frac{q_e}{2X}\, \tau \wedge e \tag{258}$$

which in Coulomb gauge leads to the usual Coulomb-potential

$$A = \frac{q_e}{\sqrt{X}}\, \mathrm{d}t \,. \tag{259}$$

The Carroll limit on the gravity side is a magnetic limit, while the Carroll limit on the Maxwell side is an electric limit, so this is an example of electric Carroll Maxwell theory coupled to magnetic Carroll gravity. It would be pleasing to see that first taking such a limit in higher dimensions and then performing a spherical reduction leads to the same 2d solutions. In [74] the combined magnetic gravity and magnetic Maxwell limits were studied for a 4d RN black hole that carries both electric and magnetic charge. It would be interesting to explore what kind of 2d model this corresponds to after spherical reduction.

Carroll extremal surfaces in the CRN geometry arise at two loci,

$$\sqrt{X}_{\pm} = m \pm \sqrt{m^2 - q_e^2} \tag{260}$$

provided the mass is positive and the charge obeys the BPS-bound

$$|q_e| \le m \,. \tag{261}$$

When saturated, $q_e^2 = m^2$, the two loci coalesce to a single degenerate Carroll extremal surface with vanishing Carroll temperature, similar to extremal Reissner–Nordström black holes.

While the CS model does not have any constant dilaton vacuum solution, the CRN model has such a solution for the value of the dilaton

$$X = q_e^2 \,. \tag{262}$$

Since this is the same value the dilaton takes at the degenerate extremal Carroll surface in the confluent case, one can interpret this constant dilaton vacuum analogously to the Robinson–Bertotti solution, i.e., as near-Carroll-extremal-surface degenerate geometry.

Generalizations of CRN to arbitrary dimension is straightforward; one just has to replace the first two terms in the potential (255) by the corresponding CST potentials corresponding to the desired dimension.

Generalizations to completely different types of charged Carroll black holes are possible as well and can be done on a case-by-case basis. Examples that come to mind are 2d type 0A string theory with equal number $q_e$ of electric and magnetic D0 branes [100, 101] and the dimensionally reduced Chern–Simons term [102, 103]. In the next Subsection, we address a different pertinent example, namely Carroll BTZ.

## 7.3   Carroll BTZ

It is not obvious how to take a Carrollian limit of the BTZ black hole [104, 105]. We take the following route. First, we Kaluza–Klein reduce along the azimuthal angle $\varphi$ to obtain the 2d Achúcarro–Ortiz model [106] and only then we take the Carroll limit.

The first step leads to a charged 2d dilaton gravity model, where the Maxwell field is the one appearing in the Kaluza–Klein ansatz ($\alpha, \beta, \gamma \in \{0, 1\}$)

$$\mathrm{d}s^2 = g_{\alpha\beta}(x^\gamma)\, \mathrm{d}x^\alpha\, \mathrm{d}x^\beta + X^2(x^\gamma)\left(\mathrm{d}\varphi + A_\alpha(x^\gamma)\, \mathrm{d}x^\alpha\right)^2 \tag{263}$$

and the associated $U(1)$ charge is the BTZ angular momentum $J$. The dimensionally reduced 2d model has the Achúcarro–Ortiz potential (see Section 6.3 in [76])

$$V_{\mathrm{AO}}(X, Y) = \frac{X}{\ell^2} - \frac{Y^2}{X^3} \tag{264}$$

where $\ell$ is the 3d AdS radius, which we set to one, $\ell = 1$. On-shell $Y = J$.

The second step consists of importing the potential (264) into generic 2d Carroll dilaton gravity (3). This yields Carroll black hole solutions we refer to as CBTZ. They are given by

$$\mathrm{d}s^2 = \mathrm{d}r^2 = \frac{\mathrm{d}X^2}{X_{\mathrm{H}}^2} \qquad\qquad v = \frac{1}{X_{\mathrm{H}}}\, \partial_t \tag{265}$$

with

$$X_{\mathrm{H}} = \pm\sqrt{X^2 - \frac{J^2}{X^2} - 2M}\,. \tag{266}$$

The solution for the gauge field follows from (248),

$$\mathrm{d}A = \frac{2J}{X^3}\, \tau \wedge e \tag{267}$$

which, in Coulomb gauge, leads to

$$A = \frac{J}{X^2}\, \mathrm{d}t\,. \tag{268}$$

As expected, there are two loci with Carroll extremal surfaces,

$$X_\pm^2 = M \pm \sqrt{M^2 - J^2} \tag{269}$$

provided the mass is positive and the angular momentum obeys the BPS bound

$$|J| \leq M\,. \tag{270}$$

When saturated, $J^2 = M^2$, the Carroll extremal surface degenerates and has vanishing Carroll temperature, similar to extremal BTZ.

In summary, the Carroll BTZ black hole (265)-(268) is the Carroll limit of the Achúcarro–Ortiz model, which in turn is a Kaluza–Klein reduction of the BTZ black hole. The Carroll BTZ black hole is a positive mass solution of 2d Carroll–Maxwell dilaton gravity (247) with the potential function (264), subject to the BPS bound (270).

# 8  Summary and Outlook

We have focused on a wide class of Carroll geometries which, despite the absence of a lightcone structure, possess black hole-like behaviour. We identified Carroll black holes as configurations exhibiting an extremal surface together with thermal properties such as finite entropy. The former is the analogue of a Lorentzian extremal surface. A crucial ingredient was incorporating the notion of Carroll thermal manifolds, introduced by relaxing the standard definition of a Carroll manifold so as to allow a vanishing "clock one-form" on isolated surfaces. Our strategy consisted of thoroughly analysing various formulations of 2d magnetic Carroll dilaton gravity models, being generic enough so as to accommodate the dimensional reduction of spherically symmetric configurations of higher-dimensional magnetic Carroll gravity. We have also shown that the processes of spherical reduction and taking the magnetic Carroll limit commute. We discussed examples in the context of magnetic Carroll gravity in diverse dimensions, including the Carroll versions of Schwarzschild, Reissner–Nordström, BTZ, as well as black hole solutions of generic Carroll dilaton gravity, including Carroll JT and Carroll Witten black holes in two spacetime dimensions. Some examples of rotating Carroll black holes were also briefly analyzed.

There are various intriguing points for further exploration:

**Mathematics of Carroll black holes** We have uncovered a couple of unusual features that could benefit from further scrutiny, specifically, the issues addressed in Subsections 2.3.3-2.3.5. Physical intuition drove us to relax the original definition of Carrollian manifolds by allowing Carrollian structure singularities, which is necessary to accommodate Carroll extremal surfaces, key protagonists in our definition of Carroll black holes. Therefore, it seems worthwhile to relax the standard mathematical notion of Carrollian manifolds and to further investigate the role of loci where the Carrollian vector field is singular, but the geometry is regular otherwise. Besides Carroll extremal surfaces, this may also include loci where the Carroll vector field tends to zero, which happens, for instance, in the limit of approaching spatial infinity from null infinity in asymptotically flat spacetimes. Having a precise (and physically relevant) definition of Carrollian singularities could open the door to further developments, such as Carrollian singularity theorems. Additionally, it is possible that such an endeavour could provide sharper or alternative definitions of Carroll extremal surfaces and Carroll black holes.

**Rotating Carroll black holes** In Section 7.3, we have presented a first example of a rotating Carroll black hole, namely Carroll BTZ. We used an intrinsic 2d approach where rotation was turned into a $U(1)$ charge after a Kaluza–Klein reduction and before taking the Carroll limit. It is natural to inquire about higher-dimensional descriptions of rotating Carroll black holes, (non-)commutativity of dimensional reduction and Carroll limit, and further questions along the lines of Fig. 1. Although (magnetic) Carroll gravity generically admits configurations with non-vanishing angular momentum [96, 98], finding higher-dimensional rotating Carroll black holes is an open task. The main obstruction comes from the Hamiltonian constraint in magnetic Carroll gravity, which requires a spatial metric with a vanishing Ricci scalar (or constant Ricci scalar, in the presence of a cosmological constant). For example, the Carrollian limit of the Kerr solution in Boyer–Lindquist or Kerr–Schild coordinates exhibits a non-vanishing spatial Ricci scalar, thus failing to satisfy the Hamiltonian constraint. It could be advantageous to seek an appropriate coordinate system that addresses this issue.

**Supersymmetric Carroll black holes** A seemingly straightforward generalization of

our work is to define and investigate Carroll supergravity (see, e.g., [107]) and supersymmetric Carroll black holes, where possibly BPS-like bounds found in Section 7 play a decisive role. Technically, we expect the simplest models to be of supersymmetric BF-type, emerging as Carrollian contractions of Lorentzian models like the super-JT model, see e.g. [108] and Refs. therein.

**Galilean black holes** There is a first order action describing 2d Galilei dilaton gravity

$$\mathcal{L} = X \, \mathrm{d}\omega + X_H \, \mathrm{d}\tau + X_P \big( \mathrm{d}e - \omega \wedge \tau \big) + \mathcal{V}_{\mathrm{Gal}}(X, X_P)\tau \wedge e \, , \tag{271}$$

which was previously considered in [43]. Given a potential analogous to the Carroll case considered in this work

$$\mathcal{V}_{\mathrm{Gal}} = -\frac{U(X)}{2} X_{\mathrm{P}}^2 + V(X) \tag{272}$$

the model is in principle solvable along the same lines. Moreover, the PSM picture allows to identify Galilean extremal surfaces as loci where $X_{\mathrm{P}} = 0$ ($X_{\mathrm{P}}$ and $X_{\mathrm{H}}$ switch roles in this case). However, these solutions cannot be assigned thermodynamic properties in the same way as in the Carroll case. One way to see this is by taking the simple example $\mathcal{V}_{\mathrm{Gal}} = X$ and partially fixing the diffeomorphism freedom such that the clock one-form is just $\tau = \mathrm{d}t$. The equations of motion then imply that $\partial_r X = 0 = \partial_r X_{\mathrm{P}}$ meaning that if there is a nontrivial configuration with $X_{\mathrm{P}} \neq 0$ the Galilei extremal surface can only lie in the future or in the past instead of at a certain radius. This makes it impossible to compactify time such that the 2d spacetime is topologically a disk and has the extremal surface in its center at the same time. This is not to say that no notion of Galilean black holes exists, it is just not as straightforward as "switching time and space". It could be interesting to see whether there is an alternative sensible way to define these objects.

**Fracton gravity** Following the relation between Carroll and particles with conserved charge and dipole momentum ("fractons") [10–13] we write down fracton BF gravity. The symmetries are spanned by $\langle H, P, Q, D \rangle$ (energy, momentum, charge, dipole moment) with the only nontrivial commutator

$$[D, P] = Q \, . \tag{273}$$

The action of fracton BF gravity is given by $I[X_I, A^I] = \frac{k}{2\pi} \int_{\mathcal{M}} \mathcal{L}$ where

$$\mathcal{L} = X_H \, \mathrm{d}A^H + X_P \, \mathrm{d}A^P + X_Q(\mathrm{d}A^Q + A^D \wedge A^P) + X_D \, \mathrm{d}A^D \, . \tag{274}$$

The Lagrange-2-form (274) corresponds to (3) upon identifying $(X_H, X_{\mathrm{P}}, X_Q, X_D)_{\mathrm{frac}} \sim (-, X_{\mathrm{P}}, X_{\mathrm{H}}, X)_{\mathrm{car}}$ and $(A^H, A^P, A^Q, A^D)_{\mathrm{frac}} \sim (-, e, \tau, \omega)_{\mathrm{car}}$, and adding the first term $X_H \, \mathrm{d}A^H$ that has no Carroll counterpart (in particular, $A^H$ is part of the geometry and not a Maxwell gauge field). While the potential in (274) is trivial, effective field theory arguments suggest it is natural to extend it to fracton dilaton gravity by adding $\mathcal{V}(X_D, X_Q) \, A^Q \wedge A^P$, since such a term is allowed by consistent deformations of the BF theory (274), see Section 7.2 of [43].

If we insist on a metric BF theory, which would be closer to Carroll/dipole Chern–Simons gravity [16, 17], we can add two nontrivial central extensions, in which case the dipole algebra admits an invariant metric [43]. This can also be generalized [43] to nontrivial cosmological constant [12, 13] or to more general gravitational models. It could be illuminating to investigate these models and their boundary conditions/actions in more detail.

**Quantum Carroll extremal surfaces** In the Lorentzian case, the concept of extremal surfaces was generalized to quantum extremal surfaces by Engelhardt and Wall [54], which, for instance, feature prominently in the island proposal [109–113]. Instead of extremizing the classical area functional, a functional that consists of the sum of area and von Neumann entropy (associated with the matter fields outside the black hole) is extremized. In quantum theories of Carroll gravity it is therefore plausible to similarly extend our notion of Carroll extremal surfaces (see Sections 4.2 and 4.3) to quantum Carroll extremal surfaces. It could be rewarding to verify whether such quantum Carroll extremal surfaces obey similar properties and theorems as in the Lorentzian case [54].

**Intrinsically higher-dimensional Gauss–Bonnet** Our main definition of Carroll temperature in Section 3.2 employs the 2d Carroll Gauss–Bonnet formula. It could be beneficial to obtain a similar result using higher-dimensional techniques, e.g., using higher-dimensional Carroll Gauss–Bonnet terms.

**Cosmology** Finally, it might be gratifying to check whether the tools we have developed in this work can be used to understand putative cosmological horizons of Carrollian cosmological geometries [19, 74, 114–116].

# Acknowledgements

We are grateful for discussions with the participants of the 1st Carroll workshop at TU Wien in February 2022 where the definition of Carroll extremal surfaces was presented for the first time. Moreover, we are grateful for discussions with participants of the 2nd Carroll workshop at UMons in September 2022, and of the workshop "Beyond Lorentzian Geometry II" at ICMS in February 2023, where some additional results from this paper were presented. In particular, we thank Jan de Boer, Laura Donnay, Adrien Fiorucci, Niels Obers, Romain Ruzziconi, Jakob Salzer, and Stefan Vandoren. DG additionally thanks Arjun Bagchi for a long-time collaboration on Carrollian physics and Dima Vassilevich for an even longer-time collaboration on 2d black holes.

**Funding information**      FE and DG were supported by the Austrian Science Fund (FWF), projects P 32581, P 33789, P 36619, and W 1252. The final part of this research was conducted while DG was visiting the Okinawa Institute of Science and Technology (OIST) through the Theoretical Sciences Visiting Program (TSVP).

JH was supported by the Royal Society University Research Fellowship Renewal "Non-Lorentzian String Theory" (grant number URF\R\221038).

SP, and in part JH, were supported by the Leverhulme Trust Research Project Grant (RPG-2019-218) "What is Non-Relativistic Quantum Gravity and is it Holographic?".

This research is partially supported by Fondecyt grants No 1211226, 1220910 (AP, RT), and 1230853 (AP).

RT thanks the support of Vicerrectoría de Investigación y Doctorados de la Universidad San Sebastián, Chile – fund "USS-FIN-23-PASI-10".

# A   Carroll symmetries

## A.1   Global Carroll symmetries

This Appendix provides a self-contained review of Carroll symmetries in any spacetime dimension $D = 1 + d$. For $d = 1$, all indices can be dropped in all formulas below (rotations do not exist in this case).

Carroll symmetries emerge as the $c \to 0$ limit of Poincaré symmetries (see [117] for some historical context). Temporal translations $H = \partial_t$, spatial translations $P_i = \partial_i$, and rotations $J_{ij} = x_i \partial_j - x_j \partial_i$ are unaffected by this limit. Thus, the only generators that change are the boosts

$$B_i = -c^2 \, t \, \partial_i - x_i \, \partial_t \qquad \overset{c \to 0}{\Rightarrow} \qquad B_i = -x_i \, \partial_t \, . \tag{275}$$

Therefore, the only commutators that change as compared to the ones in the Poincaré algebra involve Carroll boosts $B_i$:

$$[B_i, H] = 0 \qquad [B_i, B_j] = 0 \qquad [B_i, P_j] = \delta_{ij} \, H \qquad [B_k, J_{ij}] = \delta_{ki} B_j - \delta_{kj} B_i \tag{276}$$

The first commutator reveals that the Hamiltonian $H$ is a central element of the Carroll algebra, in stark contrast to the Poincaré algebra, where the Hamiltonian does not commute with Lorentzian boosts. The second commutator implies there is no Carrollian analogue of Thomas precession — two Carroll boosts always commute, regardless of the directions into which the boosts are taken. The third commutator together with the fact that $H$ commutes with the remaining Carroll generators show that $H$ can be thought of as a nontrivial central extension. The Carroll energy/mass is, therefore, an important invariant [1] quite different from the Poincaré energy. The last commutator merely shows that boosts transform as spatial co-vectors.

Finite boosts (generated by some spatial co-vector $b_i$) leave invariant space but transform time.

$$t' = t - b_i x^i \qquad\qquad x^{i\,\prime} = x^i \tag{277}$$

Thus, in Carrollian spacetimes, there is an absolute notion of space, which is the counterpart of the non-relativistic statement that in Galilean spacetimes, there is an absolute notion of time.

In the $c \to 0$ limit the Minkowski metric degenerates and acquires signature $(0, +, \dots, +)$, i.e., it becomes a purely spatial metric $h_{\mu\nu}$ with trace $d$. Its inverse (multiplied by $-c^2$) degenerates into a bi-vector $v^\mu v^\nu$ that is timelike and projects to zero with respect to the metric, $h_{\mu\nu} v^\nu = 0$. In cartesian coordinates, the Carrollian metric and vector are given by

$$\mathrm{d}s^2 = h_{\mu\nu} \, \mathrm{d}x^\mu \, \mathrm{d}x^\nu = \delta_{ij} \, \mathrm{d}x^i \, \mathrm{d}x^j \qquad\qquad v = v^\mu \, \partial_\mu = \partial_t \, . \tag{278}$$

The Carroll vector fields $\xi \in \{H, P_i, B_i, J_{ij}\}$ preserve this Carrollian structure,

$$\mathcal{L}_\xi h_{\mu\nu} = 0 = \mathcal{L}_\xi v^\mu \, . \tag{279}$$

Additionally, all "supertranslations" $\xi = f(x^i) \, \partial_t$ preserve this Carrollian structure. Thus, as opposed to Minkowski spacetimes, there are infinitely many Killing vectors. If we insist on the preservation of an invariant connection, we are led back to the original finite-dimensional Carroll symmetries [118]. A quick way to see this is to look at the infinitesimal action of a diffeomorphism generated by $\xi^\alpha(x)$ on a generic connection,

$$\delta_\xi \Gamma^\lambda{}_{\mu\nu} = \mathcal{L}_\xi \Gamma^\lambda{}_{\mu\nu} + \partial_\mu \partial_\nu \xi^\lambda \, . \tag{280}$$

The connection can therefore only be invariant if the inhomogeneous term vanishes which restricts the diffeomorphism parameter to be linear in the coordinates. In this case the supertranslations above reduce to $f(x^i) = b_i x^i + c$, reproducing (277) together with translations.

Carroll gravity can be obtained when the Carroll algebra is gauged. For details on how to gauge the Carroll algebra, see [41, 119] and the next Section.

## A.2 Local Carroll symmetries

Here we present essential aspects of local Carroll symmetries and how to relate first- and second-order formulations specialized to 1+1 dimensions. In some cases, we use the 2d Carroll dilaton gravity equations of motion for the scalar fields from the main text, see eqs. (8). Whenever we do so, we indicate this by the weakly-equal sign $\approx$.

Defining the full covariant derivative $\mathcal{D}_\mu$

$$\mathcal{D}_\mu \tau_\nu = \partial_\mu \tau_\nu - \mathbf{\Gamma}^\lambda{}_{\mu\nu}\, \tau_\lambda + \omega_\mu e_\nu \qquad\qquad \mathcal{D}_\mu e_\nu = \partial_\mu e_\nu - \mathbf{\Gamma}^\lambda{}_{\mu\nu}\, e_\lambda \tag{281}$$

and imposing the Carroll vielbein postulates

$$\mathcal{D}_\mu \tau_\nu = 0 = \mathcal{D}_\mu e_\nu \tag{282}$$

yields on-shell vanishing torsion of the affine connection

$$\mathbf{\Gamma}^\rho{}_{[\mu\nu]} = 0 \tag{283}$$

provided $\partial_H \mathcal{V} = 0$. If this is not the case, then only the spatial component of (283) vanishes ($\rho = 1$), while the temporal component ($\rho = 0$) is determined by $\partial_H \mathcal{V} \neq 0$. If $\omega$ is replaced by $\hat\omega$ the connection $\mathbf{\Gamma}^\lambda{}_{\mu\nu}$ reduces to $\Gamma^\lambda{}_{\mu\nu}$ (see Section 2.1.3).

The defining properties of the inverse vielbein are

$$v^\mu \tau_\mu = -1 \qquad\qquad v^\mu e_\mu = 0 \qquad\qquad e^\mu \tau_\mu = 0 \qquad\qquad e^\mu e_\mu = 1\,. \tag{284}$$

Under boosts they transform (off-shell) as

$$\delta_\lambda v^\mu = 0 \qquad\qquad \delta_\lambda e^\mu = -v^\mu\, \lambda \tag{285}$$

and under diffeos they transform (on-shell) with the usual Lie-derivative,

$$\delta_\xi v^\mu \approx \xi^\nu \partial_\nu v^\mu - v^\mu \partial_\nu \xi^\nu \qquad\qquad \delta_\xi e^\mu \approx \xi^\nu \partial_\nu e^\mu - e^\mu \partial_\nu \xi^\nu\,. \tag{286}$$

They are compatible with the inverse vielbein postulates.

$$\mathcal{D}_\mu v^\nu = \partial_\mu v^\nu + \mathbf{\Gamma}^\nu{}_{\mu\lambda}\, v^\lambda = 0 \qquad\qquad \mathcal{D}_\mu e^\nu = \partial_\mu e^\nu + \mathbf{\Gamma}^\nu{}_{\mu\lambda} e^\lambda + v^\nu \omega_\mu = 0 \tag{287}$$

Defining the usual covariant derivative $\mathbf{\nabla}_\mu$ in terms of the affine connection $\mathbf{\Gamma}^\lambda{}_{\mu\nu}$ yields the compatibility conditions

$$\mathbf{\nabla}_\mu v^\nu = 0 \qquad\qquad \mathbf{\nabla}_\mu g_{\nu\lambda} = 0 \tag{288}$$

where we defined the spatial metric as bilinear in the spatial vielbein

$$g_{\mu\nu} = e_\mu e_\nu\,. \tag{289}$$

A metric with upper indices is similarly defined.

$$g^{\mu\nu} = e^\mu e^\nu \tag{290}$$

Since the metric with lower indices is Carroll boost invariant,

$$\delta_\lambda g_{\mu\nu} = 0 \tag{291}$$

together with the Carroll boost invariant vector field $v^\mu$ it defines a meaningful (i.e., boost invariant) notion of Carrollian geometry. In the context of 2d Carroll dilaton gravity, one should also consider the dilaton as part of the geometry, which is possible since the dilaton is also Carroll boost invariant.

Defining the usual Riemann tensor

$$\left[\boldsymbol{\nabla}_\mu, \boldsymbol{\nabla}_\nu\right] k^\lambda = \mathbf{R}^\lambda{}_{\rho\mu\nu} k^\rho - 2\boldsymbol{\Gamma}^\rho{}_{[\mu\nu]} \boldsymbol{\nabla}_\rho k^\lambda \tag{292}$$

relates it through the vielbein postulates to the Carrollian first-order variables.

$$\mathbf{R}^\lambda{}_{\rho\mu\nu} = -v^\lambda e_\rho\big(\partial_\mu\omega_\nu - \partial_\nu\omega_\mu\big) \tag{293}$$

Note that there is no bilinear term in the connection since we are in 2d. Similarly to the behaviour of the connection, using $\hat\omega$ instead of $\omega$ in this expression reduces $\mathbf{R}^\lambda{}_{\rho\mu\nu}$ to the Carrollian curvature tensor $R^\lambda{}_{\rho\mu\nu}$ as used in the main part. We get as only non-vanishing components

$$\mathbf{R}^t{}_{rtr} = -\mathbf{R}^t{}_{rrt} = -\partial_X \mathcal{V}(X, X_{\mathrm{H}}) \tag{294}$$

According to the general analysis of [41], the Carrollian affine connection in our case is given by

$$\boldsymbol{\Gamma}^\lambda{}_{\mu\nu} = -v^\lambda\big(\partial_\mu\tau_\nu + \omega_\mu e_\nu\big) + e^\lambda\partial_\mu e_\nu \, . \tag{295}$$

This result is compatible with torsion

$$T^\lambda{}_{\mu\nu} = \boldsymbol{\Gamma}^\lambda{}_{[\mu\nu]} \qquad \Rightarrow \qquad T^\lambda{}_{\mu\nu} e_\lambda \approx 0 \qquad T^\lambda{}_{\mu\nu} \tau_\lambda \approx -\partial_{\mathrm{H}}\mathcal{V}(X, X_{\mathrm{H}})\, \tau_{[\mu} e_{\nu]} \tag{296}$$

which vanishes on-shell if and only if the potential is $X_{\mathrm{H}}$-independent, $\partial_{\mathrm{H}}\mathcal{V}(X, X_{\mathrm{H}}) = 0$, see (8b). The Riemann tensor can be computed as well, matching the result above.

$$\mathbf{R}^\lambda{}_{\rho\mu\nu} = \partial_\mu\boldsymbol{\Gamma}^\lambda{}_{\nu\rho} + \boldsymbol{\Gamma}^\lambda{}_{\mu\sigma}\boldsymbol{\Gamma}^\sigma{}_{\nu\rho} - \big(\mu \leftrightarrow \nu\big) = -v^\lambda e_\rho\big(\partial_\mu\omega_\nu - \partial_\nu\omega_\mu\big) \tag{297}$$

# B   Lorentzian and Carrollian PSMs

The purpose of this appendix is to show that there is a target space diffeomorphism that maps Lorentzian PSMs to Carrollian PSMs. For a more detailed explanation of the connection between PSMs and Lorentzian 2d dilaton gravity we refer to [64]. The application of target space diffeomorphisms in the Lorentzian case was elaborated on in [120, 121]. The general form of a PSM is

$$I_{\mathrm{PSM}}[A_I, X^I] = \frac{k}{2\pi} \int_{\mathcal{M}} \left(X^I \, \mathrm{d}A_I + \frac{1}{2}\, P^{IJ}(X^K)\, A_I \wedge A_J\right) . \tag{298}$$

It describes Lorentzian 2d dilaton gravity if a 3d target space coordinatized by $X, X^+, X^-$ as well as a Poisson tensor of the form

$$P^{IJ} = \begin{pmatrix} 0 & -X^+ & X^- \\ X^+ & 0 & \hat{\mathcal{V}}(X, X^+X^-) \\ -X^- & -\hat{\mathcal{V}}(X, X^+X^-) & 0 \end{pmatrix} \tag{299}$$

are chosen. The connection to gravitational variables is achieved by a background target space metric

$$\eta^{IJ} = \begin{pmatrix} 0 & 0 & 0 \\ 0 & 0 & 1 \\ 0 & 1 & 0 \end{pmatrix} \tag{300}$$

that allows constructing the 2d worldsheet metric from the PSM connection,

$$g_{\mu\nu} = \eta^{IJ} A_{I\,\mu} A_{J\,\nu} = e_\mu^+ e_\nu^- + e_\mu^- e_\nu^+ \ . \tag{301}$$

The Lorentzian Poisson tensor (299) can now be mapped to the Carrollian Poisson tensor (27) by the target space diffeomorphism[14]

$$X_{\text{H}} = \sqrt{2X^+ X^-} \qquad\qquad X_{\text{P}} = \frac{X_{\text{H}}}{2} \ln \frac{X^-}{X^+} \ . \tag{302}$$

Explicit calculation of the transformed Poisson tensor components yields

$$P^{X_{\text{H}}} = P^{X+} \frac{\partial X_{\text{H}}}{\partial X^+} + P^{X-} \frac{\partial X_{\text{H}}}{\partial X^-} = 0 \tag{303}$$

$$P^{X_{\text{P}}} = P^{X+} \frac{\partial X_{\text{P}}}{\partial X^+} + P^{X-} \frac{\partial X_{\text{P}}}{\partial X^-} = X_{\text{H}} \tag{304}$$

$$P^{\text{HP}} = P^{+-} \left( \frac{\partial X_{\text{H}}}{\partial X^+} \frac{\partial X_{\text{P}}}{\partial X^-} - \frac{\partial X_{\text{H}}}{\partial X^-} \frac{\partial X_{\text{P}}}{\partial X^+} \right) = \hat{\mathcal{V}}(X, \tfrac{1}{2} X_{\text{H}}^2) \ . \tag{305}$$

With the identification $\hat{\mathcal{V}}(X, \tfrac{1}{2} X_{\text{H}}^2) = \mathcal{V}(X, X_{\text{H}})$ this produces indeed the Carrollian Poisson tensor (27).

Besides changing the Poisson tensor, we also need to change the map from PSM to worldsheet geometry variables, which in the Lorentzian case is given by the background target space metric (300). Taking the Carrollian limit thereof yields

$$\eta_{\text{C}}^{IJ} = \begin{pmatrix} 0 & 0 & 0 \\ 0 & 0 & 0 \\ 0 & 0 & 1 \end{pmatrix} \tag{306}$$

so that the worldsheet metric degenerates, as required.

$$h_{\mu\nu} = \eta_{\text{C}}^{IJ} A_{I\,\mu} A_{J\,\nu} = e_\mu e_\nu \tag{307}$$

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
