# Peer review of "Carroll black holes"

_SciPost Physics_

## Round 1 · Referee Report · Anonymous (Referee 1) · 2023-10-7

Strengths

1-The work emphasizes the importance of 2-d dilaton gravity, as well as its Carrollian limit, particularly in the context of higher-dimensional spherically symmetric black holes.
2-The theoretical tools developed in Sections 2-4 are comprehensively applied to black hole scenarios that are well known in the literature and of interest to the broader gravitational community.

Weaknesses

1-The analysis developed in this work is not able to address non-spherically symmetric black holes in higher-than-three spacetime dimensions (e.g. the Kerr black hole).

Report

The authors introduce the notion of Carroll black holes, via a systematic study of the 2-d Carroll dilaton gravity. After presenting the 2-d Carroll dilaton gravity theory in various formalisms and emphasizing its relevance for black hole geometries in the broader context, they construct its solution space.

By studying singularities in the Carrollian structure, they are able to identify Carrollian versions of various thermal properties. This further allows them to introduce the new notion of "Carroll extremal surfaces" which is paramount to their definition of "Carroll black holes".

This work provides a substantial conceptual advancement in the understanding of black hole physics and their connection to Carroll gravity, and is clearly written. I find the content of the paper to meet the journal's acceptance criteria, after being supplemented with a more comprehensive summary of the results in the concluding remarks. I therefore recommend this paper to be published in SciPost Physics.

Requested changes

1-Supplement the work with a component that summarizes the results before addressing future directions, e.g. in the form of a "Summary and Outlook" section.

---

## Round 1 · Referee Report · Anonymous (Referee 2) · 2023-10-8

Strengths

1- A novel black hole definition that works for Carrollian geometry and presumably beyond.

2- A thorough discussion of how these "black holes" arise in Carrollian dilaton gravity.

Weaknesses

1- Except for the Carroll-Schwarzschild black hole, which has been discussed elsewhere (e.g. in refs. [96] and [74], the latter of which includes a discussion of the entropy), the treatment is limited to 2d. In particular, the definition of a Carrollian extremal surface involves the dilaton.

2- A primary motivation is a better definition of black holes. A small discussion of how (and whether) the definition presented extends to Galilean and Aristotelian geometries would be helpful.

Report

This paper introduces a notion of Carrollian "black holes", whose defining characteristic, in the absence of horizons familiar from ordinary black holes, is a combination of a Carrollian extremal surface and a set of thermal properties. These thermal properties require that the Carrollian structure be allowed to contain isolated singular points. To illustrate the existence of such solutions, the authors consider (magnetic) Carrollian dilaton gravity, which in particular captures the spherically reduced 4d Carroll-Schwarzschild black hole. In addition, a number of 2d solutions are discussed, such as the Carrollian analogs of the BTZ black hole and the Witten black hole.

These results are novel, interesting and timely. The paper meets the expectations of SciPost Physics and, up to a number of minor points, I recommend this article for publication.

Requested changes

Overall remarks: 1- As the authors themselves more or less point out, there is nothing "black" about Carrollian black holes. To avoid a slightly misleading title, perhaps it would be more appropriate to title the article "Carrollian analogs of black holes".

2- It would be useful to have a 4d perspective on charged and rotating BHs that go beyond the results in sec. 7. As far as I can tell, it is straightforward to expand the Kerr and RN BHs in powers of $c^2$. Would the resulting Carrollian geometries qualify as "Carrollian black holes"?

3- Related to point (2) in "weaknesses" above, a small discussion of how this definition would apply in Galilean and Aristotelian contexts would improve the paper.

Punctual remarks: 1- Starting on page 3, the limit considered is termed the "ultrarelativistic" limit. However, the ultrarelativistic limit is $v/c\rightarrow 1$ rather than $v/c\rightarrow 0$, which is the ultralocal limit. This should be changed throughout.

2- Below eq. (3), $\Theta$ is referred to as the intrinsic torsion. However, the intrinsic torsion of a Carrollian structure is symmetric in general dimension. The relation between $\Theta$ and the intrinsic torsion in 2d is given in-line below eq. (21).

3- The Carroll boost parameter $\lambda$ in eq. (4) has an unconventional sign. Why?

4- It would be useful to include the completeness relation $\delta^\mu_\nu = -v^\mu\tau_\nu + h^{\mu\rho}h_{\rho\nu}$ in eq. (17).

5- Perhaps switch eqs. (26) and (27). This would help define the indices $I,J,\dots$.

6- The Poisson tensor in (26) does not appear to be antisymmetric. Why not?

7- To avoid a proliferation of terminology, the term "pre-Carrollian", which first appears above eq. (38), should be replaced with "pre-ultralocal (PUL)" throughout.

8- Perhaps include a spoiler below eq. (49) that reveals that $\gamma$ ultimately will not play a role.

9- In eqs. (64) and (66) the identity $\overset{(C)}{\nabla}_\mu V^\mu = -\mathcal{K}$ is used. Perhaps include this identity as an aid to the reader.

10- The second equality of eq. (70) is a "weak" equality in the sense that is is only valid on-shell. It should probably be replaced with an "$\approx$".

11- In sec. 3.1, there is a recurring typo: $\partial M$ should be replaced with $\partial \mathcal{M}$.

12- Above eq. (199): "Lagrangean" should be "Lagrangian".

13- Explicitly state that this is the spherically reduced Carroll-Schwarzschild black hole above eq. (211).

14- What does "Minkowski coordinates" mean above eq. (276)? Isn't the point rather that there exists Cartesian coordinates in flat space such that the Carrollian structure takes the form of eq. (276)?

15- Below eq. (277): the fact that the infinitely many Killing vectors of a flat Carrollian structure reduce to the Carrollian symmetries was first discussed in arXiv:1402.0657, which should be included as a reference. It would be nice, but not necessary, to explicitly demonstrate the statement which is frequently made but seldomly shown.

16- Appendix B could be made more self-contained by, for example, explicitly including eq. (25) and by including references. I recommend a small rewriting of this appendix.

---

## Editorial Decision

resubmitted